# Multi-omics characterization of the monkeypox virus infection

Yiqi Huang [1,13], Valter Bergant [1,13], Vincent Grass [1,13], Quirin Emslander[1,13], M. Sabri Hamad [1], Philipp Hubel[2,3], Julia Mergner [4], Antonio Piras[1], Karsten Krey [1], Alexander Henrici[1], Rupert Öllinger [5], Yonas M. Tesfamariam [6], Ilaria Dalla Rosa[7], Till Bunse [1], Gerd Sutter[8,9], Gregor Ebert [10], Florian I. Schmidt [6], Michael Way [7,11], Roland Rad [5], Andrew G. Bowie [12], Ulrike Protzer[9,10] & Andreas Pichlmair [1,9] ✉

Multiple omics analyzes of Vaccinia virus (VACV) infection have defined molecular characteristics of poxvirus biology. However, little is known about the monkeypox (mpox) virus (MPXV) in humans, which has a different disease manifestation despite its high sequence similarity to VACV. Here, we perform an in-depth multi-omics analysis of the transcriptome, proteome, and phosphoproteome signatures of MPXV-infected primary human fibroblasts to gain insights into the virus-host interplay. In addition to expected perturbations of immune-related pathways, we uncover regulation of the HIPPO and TGF-β pathways. We identify dynamic phosphorylation of both host and viral proteins, which suggests that MAPKs are key regulators of differential phosphorylation in MPXV-infected cells. Among the viral proteins, we find dynamic phosphorylation of H5 that influenced the binding of H5 to dsDNA. Our extensive dataset highlights signaling events and hotspots perturbed by MPXV, extending the current knowledge on poxviruses. We use integrated pathway analysis and drug-target prediction approaches to identify potential drug targets that affect virus growth. Functionally, we exemplify the utility of this approach by identifying inhibitors of MTOR, CHUK/IKBKB, and splicing factor kinases with potent antiviral efficacy against MPXV and VACV.

Monkeypox (mpox) virus (MPXV) is a pathogenic orthopoxvirus and etiological agent of the zoonosis mpox, first identified in 1970[1]. The virus species is separated into clade I (Central African, Congo basin) and clade II (West African), the latter of which includes the 2022 mpox outbreak strain that caused the first widespread community transmission of MPXV outside Africa[2–4]. Its rapid spread in 2022 resulted in the WHO's declaration of a Public Health Emergency of International Concern (PHEIC) and has so far caused over 88,000 cases in 110 countries

[1]Institute of Virology, Technical University of Munich, School of Medicine, Munich, Germany. [2]Innate Immunity Laboratory, Max-Planck Institute of Biochemistry, Munich, Germany. [3]Core Facility Hohenheim, Universität Hohenheim, Stuttgart, Germany. [4]Bavarian Center for Biomolecular Mass Spectrometry at University Hospital rechts der Isar (BayBioMS@MRI), Technical University of Munich, Munich, Germany. [5]Institute of Molecular Oncology and Functional Genomics and Department of Medicine II, School of Medicine, Technical University of Munich, Munich, Germany. [6]Institute of Innate Immunity, Medical Faculty, University of Bonn, Bonn, Germany. [7]Cellular signalling and cytoskeletal function laboratory, The Francis Crick Institute, 1 Midland Road, London NW1 1AT, UK. [8]Institute for Infectious Diseases and Zoonoses, Department of Veterinary Sciences, LMU Munich, Munich, Germany. [9]German Centre for Infection Research (DZIF), Partner site Munich, Munich, Germany. [10]Institute of Virology, Technical University of Munich, School of Medicine/Helmholtz Munich, Munich, Germany. [11]Department of Infectious Disease, Imperial College, London, UK. [12]School of Biochemistry and Immunology, Trinity Biomedical Sciences Institute, Trinity College Dublin, Dublin, Ireland. [13]These authors contributed equally: Yiqi Huang, Valter Bergant, Vincent Grass, Quirin Emslander. ✉e-mail: andreas.pichlmair@tum.de

worldwide (CDC, 27th July 2023). Even though MPXV and vaccinia virus (VACV), a well-studied poxvirus that served as a vaccine against smallpox, have a high sequence similarity[5] (Supplementary data 1), their pathology in humans differs. While VACV mostly causes cutaneous lesions limited to the site of inoculation[6], Mpox can cause a severe and, in some cases, disfiguring disease associated with non-specific prodromal symptoms - skin lesions, fever, fatigue and headache - followed by a distinctive pox-like rash[7–9] and mpox-specific lymphadenopathy[10]. There is currently no approved treatment specifically for mpox. Cross-protective smallpox vaccines have been shown to be active against mpox both prophylactically and therapeutically[11–14]. However, due to declining immunity owing to the end of routine smallpox vaccination, poxviruses represent a re-emerging threat to society and individuals[15]. Anti-vaccinia virus antibodies[16], as well as antivirals tecovirimat (TPOXX)[17] and cidofovir/brincidofovir[18,19] have been suggested for clinical use due to their effectiveness against other orthopoxvirus infections. However, despite several ongoing clinical trials for these treatment options against mpox, data on the efficacy of these treatment options remains pending. Furthermore, treatment parameters such as dose and treatment duration need to be established, and the potential emergence of drug-resistant viruses must be monitored[20].

The global spread of MPXV has demonstrated the need for a better understanding of the mpox disease, particularly with respect to other poxviruses. In the past, multi-omics systems analyzes of viruses like ZIKV and SARS-CoV-2 have greatly increased our knowledge of the epidemic or pandemic viruses and facilitated drug development[21–25]. Such information also forms a central pillar of global pandemic preparedness programs[26]. In particular, multiple omics studies on VACV have helped us to understand the fundamental principles of poxvirus pathophysiology[27–30] and revealed the influence of VACV on central biological processes such as cell cycle[31], cell death[32], immunomodulation[33] and host shut-off[34]. However, despite a limited number of transcriptomic studies[35,36] and one plasma proteomic study of MPXV-infected patients[37], there is a lack of multi-omic studies of MPXV infection in a coherent system. Here, we report a time-resolved multi-omics study of MPXV infection in primary human cells, including transcriptomics, proteomics, and phosphoproteomics. Our extensive dataset highlights multiple signaling events and hotspots that are perturbed during MPXV infection, thus improving our understanding of viral biology and helping to guide the rational design of both virus- and host-directed therapies.

## Results

### Multi-omics profiling of MPXV-infected primary cells

To explore the cellular responses of primary human foreskin fibroblasts (HFFs) to MPXV infection, we profiled the effects of viral infection on the transcriptome, proteome, and phosphoproteome in a time-resolved manner (Fig. 1a). Transcriptomic profiling allowed us to quantify the expression of 12970 host genes, 1827 of which were differentially regulated during MPXV infection, with the highest degree of transcriptional dynamics at the late infection stage (Fig. 1b, Supp. Fig. 1a–c, Supplementary data 2). In agreement with previous transcriptomic studies from VACV infection[38–40], we observed rapid and potent up-regulation of multiple normally non-polyadenylated RNA species (e.g., histones, U1, and 7SK) in our polyA-specific sequencing data (Fig. 1c, Supplementary data 2), which may be attributed to the activity of the viral poly(A) polymerase VP55(F1, Cop-E1)/VP39(L3, Cop-J3)[41]. We observed multiple host kinases relevant for poxvirus infection regulated at the transcriptomics level, exemplified by the Aurora (*AURKA* and *AURKB*) and Polo-like (*PLK1* and *PLK2*) kinases[42]. Moreover, we detected a marked upregulation of the AP-1 transcription factor components (e.g., *JUN, JUNB, FOS*) - MAPK/AP-1 pathway plays a prominent role in poxvirus infection and shapes the inflammatory responses to the infection[43]. In line with these expression profiles, MPXV activated a pro-inflammatory signature as exemplified by the

significant upregulation of cytokines and chemokines (e.g., *CXCL1, IL8 (CXCL8), IL6,* and *IL11*) (Fig. 1c, Supp. Fig. 1a–c). In contrast to the inflammatory pattern, the cellular interferon response was tightly repressed with no signs of induction of interferons or canonical ISGs, demonstrating prominent engagement of viral regulatory processes to impair antiviral immunity (Fig. 1c).

Perturbations of translation and proteostasis are common features among pathogenic viruses, including poxviruses[34]. Proteomics profiling allowed us to quantify 7701 proteins, 1216 of which were differentially regulated during MPXV infection (Fig. 1d, Supplementary data. 3). Compared to similar datasets based on profiling VACV[28] and MVA[30] infections in fibroblasts, our analysis identified substantially more dysregulated cellular proteins (Fig. 1d). In agreement with these studies, we observed a prominent trend toward the downregulation of host proteins after virus infection (Fig. 1d, Supp. Fig. 1d–f). Intersecting the datasets identified a small core set of proteins that were affected by all poxviruses (Fig. 1d). Among them were collagens and many other proteins involved in tissue homeostasis, such as MMPs and TIMPs, which may contribute to the perturbation of tissue homeostasis and induction of tissue damage[28] (Fig. 1e, Supp. Fig. 1d–f, Supplementary data 3). Interestingly, we found that the mRNA levels of MMPs and TIMPs were mostly unchanged, while their protein abundance was reduced during MPXV infection (Fig. 1c, e; Supplementary data 3). We validated the observed relationships between RNA and protein levels in independent experiments. An increase in virus transcripts (Fig. 1h) and viral proteins (Fig. 1i, j) confirmed the successful infection of HFFs. In line with omics measurements, we validated the discrepancy between abundances of MMP14 mRNA and protein in infected cells (Fig. 1h–j). Within our MPXV dataset, we found consistent regulation of proteins across the course of infection (Fig. 1e, Supp. Fig. 1d–f) and, again, tight control of type I interferon response (Fig. 1e). Particularly evident was the absence of IFIT proteins, as compared to IFIT transcripts (Supplementary data 3). IFIT proteins are directed towards proteasomal degradation by poxvirus ubiquitin ligases such as the VACV protein Cop-C9[44]. In line with this, our in silico comparison of the benchmark VACV proteome (Proteome ID: UP000000344) and the MPXV proteome (GenBank ID: ON563414.3) reveals that there are at least two conserved C9 homologs in MPXV (Supplementary data 1). Additional evidence for a dysregulated immune response came from the upregulation of PTGS2 (COX2), a key enzyme for prostaglandin synthesis, which may explain the previously reported increase in prostaglandin production upon poxvirus infection[45]. We validated the simultaneous regulation of PTGS2 on both mRNA and protein levels (Fig. 1h–j). Along with the regulation of inflammatory processes, the cysteine protease CASP1, involved in inflammasome response and pyroptosis induction[46], was strongly downregulated at all analyzed time points, suggesting regulation of cell death and inflammatory pathways. Additionally, downregulation of BNIP3L (NIX), a mitochondrial pro-apoptotic Bcl-2 family member[47], previously reported to interact with viral and cellular anti-apoptotic proteins[48], could be observed. Furthermore, the abundance of lamin-A/C, the structural proteins of the nucleus, was not consistently affected by MPXV infection despite a profound downregulation of its mRNA (Fig. 1h–j). Surprisingly, we detected a prominent downregulation of AKT kinases AKT1, AKT2, and AKT3 (Fig. 1e), which was unexpected since poxviruses are known to rely on the PI3K/AKT pathway at multiple points in the virus life cycle[49]. In addition, we identified a significant downregulation of multiple proteins associated with TGF-β signaling (e.g., THBS1, FN1, and TGFBI) (Fig. 1e, Supp. Fig. 1d–f), a prominent pathway that is also involved in tissue homeostasis. We confirmed that THBS1 is regulated at the mRNA level (Fig. 1h–j). Taken together, the proteomics profiling suggests there is extensive regulation of communication between infected cells and the surrounding tissues.

The differential expression of central kinases suggested that MPXV profoundly perturbs cellular signaling. Indeed, time-resolved

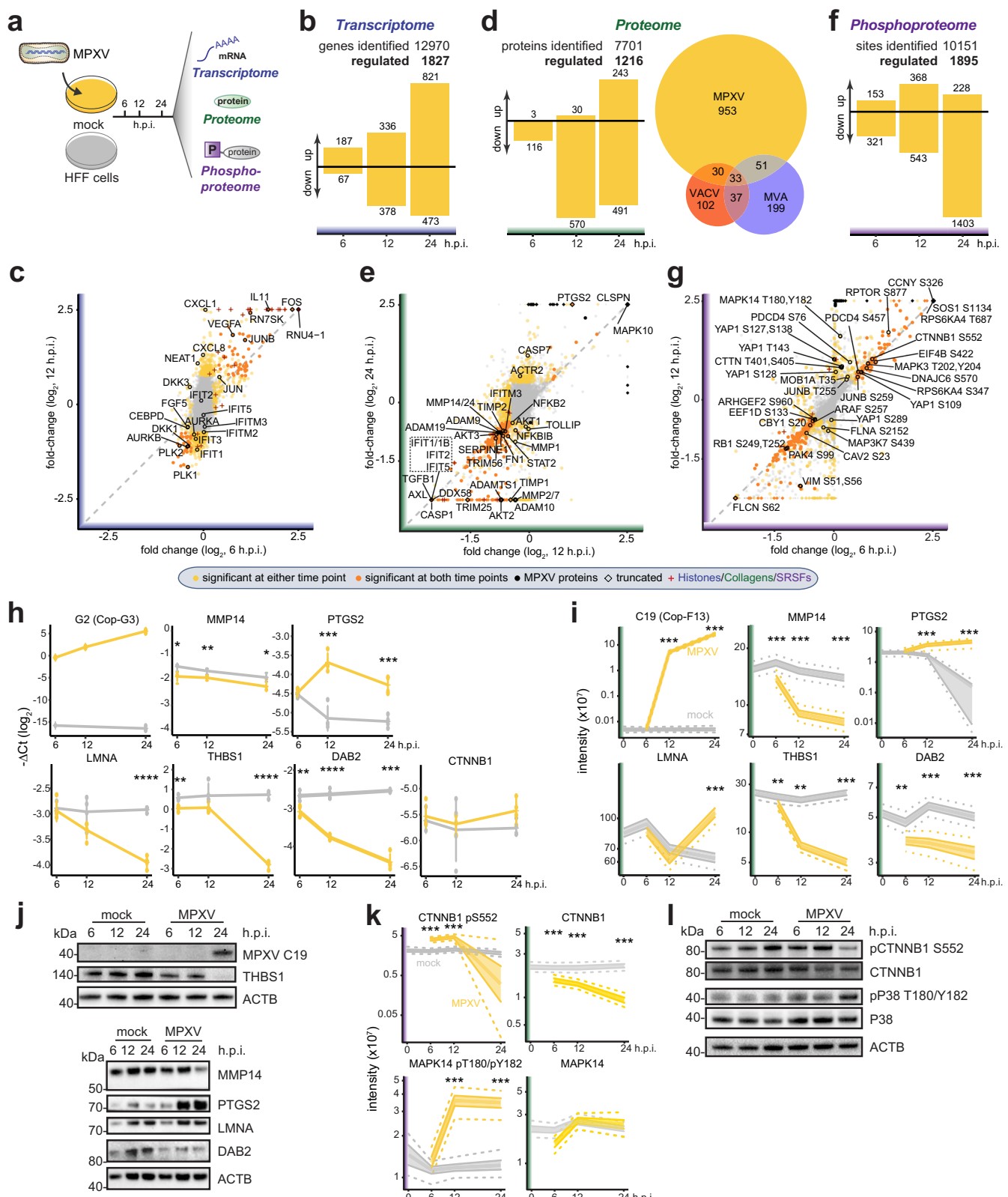

phosphoproteomics analysis of MPXV-infected HFF cells identified 10151 phosphosites, of which 1895 were significantly changed by the infection. Notably, our study identifies 901 phosphosites on host proteins that have not been described so far (Phosphositeplus (v2022.08)), which underlines the high quality of the dataset (Fig. 1f, Supp. Fig. 1g–i, Supplementary data 4). We observed up and down-regulation of phosphopeptides early after infection, which was

followed by a major shift towards downregulation at the late times (Fig. 1f, Supp. Fig. 1g-i). The differential phosphorylation of proteins was especially prominent at 24 hours post-infection and may be explained by the late expression profiles of the viral kinases (C16 (Cop-F10) and B3 (Cop-B1)) as well as the viral phosphatase H1 (Cop-H1). A high degree of consistency between differentially regulated phosphosites across the course of infection suggests that MPXV infection

**Fig. 1 | Multi-omics analysis of MPXV-infected primary human foreskin fibroblasts. a**–**g** Transcriptomes, proteomes, and phosphoproteomes of HFFs infected with MPXV (MOI 3) were profiled at 0, 6, 12, or 24 hours post-infection (h.p.i.). Bayesian linear modeling was used to determine the statistical significance of observed changes on distinct omics layers relative to time-matched mock controls. **a** Schematic representation of the multi-omics profiling of the MPXV-infected primary HFFs. **b**, **d**, **f** Numbers of significantly changed host transcripts **b**, proteins **d**, or phosphosites **f** at the indicated times after MPXV infection. In **d**, Euler diagram shows the number proteins significantly changed by MPXV (this study), VACV[28], and MVA[30] infection. **c**, **e**, **g** Scatterplots depicting fold-changes of the abundance of transcripts **c**, proteins **e**, or phosphosites **g** between MPXV-infected HFFs and timepoint-matched mock controls. Statistically significant events are yellow (at either time point) or orange (at both time points), viral transcripts, proteins, or phosphosites are black. Histones **c**, collagens **e**, or Serine and arginine Rich Splicing Factors (SRSFs) **g** are crosses. Diamonds: log2 fold change was truncated to fit into the plot. **h** Expression levels, as measured by RT−qPCR, of MPXV G2 and host

transcripts relative to RPLP0 in MPXV-infected (MOI 3) HFFs at indicated time points. Error bars: mean and standard deviation (Two-sided Welch's t-test, unadjusted p-value, n = 4 independent experiments). **i** Abundances of MPXV protein C19 and human proteins as determined by proteomic analysis of MPXV-infected HFFs. **j** Expression of MPXV protein C19 and human proteins in MPXV-infected HFFs (n = 3 independent experiments). **k** Abundance of CTNNB1 phospho-S552 and MAPK14 phospho-T180/Y182 and abundance of CTNNB1 and MAPK14 proteins as determined by proteomics analysis. **l** Representative western blots showing abundance changes of the phosphorylated and total CTNNB1 and P38 (MAPK11-14) in MPXV-infected HFFs (n = 3 independent experiments). For **i** and **k**, the line indicated the modeled median, and the shaded region and the dotted line represented 50% and 95% credible intervals, respectively (n = 5 independent experiments). Bayesian linear model-based unadjusted two-sided p-value: *: p-value ≤ 0.05; **: p-value ≤ 0.01; ***: p-value ≤ 0.001; ****: p-value ≤ 0.0001. Source data are provided as a Source Data file.

tightly regulates a subset of phosphosites (Supplementary data 4). We observed widespread dephosphorylation of serine and arginine-rich splicing factors (SR proteins, e.g. SRSF1 pS234, SRSF10 pS133, SRSF6 pS303, SRSF9 pS211/S216) putatively caused by H1 (Cop-H1), which may collectively impair the host RNA splicing[50] (Supp. Fig. 1g–i). We also identified other known phosphosites with annotated functions[51] that were differentially regulated upon MPXV infection. For example, MPXV infection resulted in differential phosphorylation of proteins involved in cytoskeleton organization (e.g., CTTN and VIM), mTOR regulation (AKT1S1, FLCN, and RPTOR), regulation of cell survival and cell death (ACIN1, BCLAF1, PAK2, and PAK4), HIPPO-YAP pathway regulation (YAP1, LATS1, MOB1A, and NF2), translation control (EEF1D, EIF4B, EIF4G2, and PDCD4), MAPK signaling (ARAF, MAPK14, MAP3K7, MAPK3 and SOS1), and WNT/beta-catenin pathway (AMOTL1, CBY1, CSNK1E and CTNNB1) (Fig. 1g, Supp. Fig. 1g–i). In particular, we validated CTNNB1 S552 and p38 (MAPK11-14) T180/Y182 phosphorylation as well as CTNNB1 and p38 protein abundance dynamics upon MPXV infection (Fig. 1h, k, l). The observed activation of Wnt and MAPK signaling may arise from the downregulation of DAB2, a previously described inhibitor of Wnt and MAPK pathways[52,53], which we validated by western blotting (Fig. 1h–j). Notably, a number of proteins exhibited either stabilizing or destabilizing phosphorylation and concomitant changes in protein levels (e.g., YAP1 and HSP1A1, Supp. Fig. 1j). Overall, the transcriptomics, proteomics and phosphoproteomics analyzes indicated MPXV-dependent regulation of central cellular kinases such as cyclin-dependent kinases, cAMP-dependent PKA, MAPKs and AKT kinase, which are linked to key biological processes such as cell cycle progression, survival, growth, metabolism and cell morphology and motility.

## Viral protein dynamics

Beyond host perturbations, we could identify 161 MPXV proteins with distinct temporal expression patterns in our dataset. Uniform manifold approximation and projection (UMAP) of modeled protein abundances allowed the classification of viral proteins into three groups - proteins first detected after 6, 12, or 24 hours post-infection (Fig. 2a, Supplementary data 5). The members of the 7-protein complex (7PC), which is required for membrane re-organization during viral assembly, were only detectable after 12 h.p.i. as compared to major virion structural proteins that are detectable starting from 6 h.p.i. The expression patterns of, e.g., the viral NF-κB inhibitor O2 (Cop-M2), viral phosphatase H1 (Cop-H1) and viral kinase C16 (Cop-F10) matched the previous reports for other poxviruses[28,54] (Fig. 2b). Moreover, we identified 127 phosphosites on viral proteins, of which 66 have not been observed in previous studies on VACV[27,55] (Supp. Fig. 2a, Supplementary data 4). Amongst them were 12 phosphosites that were unique to MPXV and do not exist in VACV, and 3 were found on B21 that does not have a homolog in VACV (Supp. Fig. 2b).

Interestingly, more close investigation showed that a subset of homologous MPXV and VACV proteins are phosphorylated at similar phosphosites (e.g. C23 (Cop-F17)), indicating engagement of conserved regulatory processes. For some homologous proteins, we identified sites that have not been detected in other studies (Supp. Fig. 2c), which allows additional considerations on the structural properties of these proteins and their regulation. Employing UMAP, we classified all MPXV phosphorylation sites into four groups, depending on their phosphorylation kinetics and abundance (Fig. 2c, Supplementary data 5). This analysis allowed us to classify phosphosites according to their differential phosphorylation patterns, exemplified by the envelope phosphoglycoprotein A35 pS172 (Cop-A33, 6 h.p.i.) and membrane protein A14 pS40 (Cop-A13, 12 h.p.i.) and pS35 (24 h.p.i., Fig. 2d,e). Some phosphosites on host proteins were previously reported to be influenced by the viral kinase C16 (Cop-F10), for instance, phosphorylation of mDia/DIAPH1 and dephosphorylation of ARHGAP17 through an indirect mechanism[56] (Supp. Fig. 2d). Both proteins contribute to cytoskeletal reorganization, which is a hallmark of poxvirus infection[57]. Viral kinase B3 (Cop-B1) is also known to phosphorylate host proteins, such as RACK1 (Supp. Fig. 2e), to modulate the translation of host and viral mRNA[58].

The phosphorylation patterns observed in viral proteins allow us to predict the involvement of putative host kinases. Interestingly, many viral phosphosites were positioned in motifs that could potentially be phosphorylated by host kinases (Fig. 2e, Supplementary data 5). While some of the identified motifs were highly promiscuous and could be targeted by numerous kinases (e.g., Q1 S338, A13 T97, B21 T682, C23 S61), the majority of identified motifs were selective for a more limited set of kinases (Supp. Fig. 2f). Searching for kinases that could potentially phosphorylate viral proteins highlighted the role of host kinase recognition motifs enriched at the viral proteins' phosphosites (Fig. 2f). In particular, multiple MAPK motifs were enriched, and many phosphomotifs of MPXV proteins were predicted to be phosphorylated by MAPKs (Supp. Fig. 2g). Q1 of MPXV is the most prominent example, where its 3 phosphomotifs were predicted to be phosphorylated by more than 20 MAPKs. Collectively, the highly dynamic phosphorylation of viral proteins in conjunction with the enrichment of host kinase recognition motifs prompted us to explore the phosphorylation patterns of specific viral proteins in more detail. Notably, we found multiple phosphosites on individual viral proteins with different temporal kinetics (Fig. 2e), such as on H5 (Cop-H5) (Fig. 2g) and A14 (Cop-A13) (Supp. Fig. 2h). With nine identified phosphosites that located into three distinct clusters, H5 is the most phospho-decorated viral protein (Fig. 2e), potentially reflecting its multifunctional nature and essential involvement in virus replication and virion maturation[59]. VACV H5 is a multimeric protein that can bind to double-stranded (ds) DNA[60], and we confirmed that MPXV H5 similarly self-associates and binds to dsDNA (Supp. Fig. 2i). In MPXV-

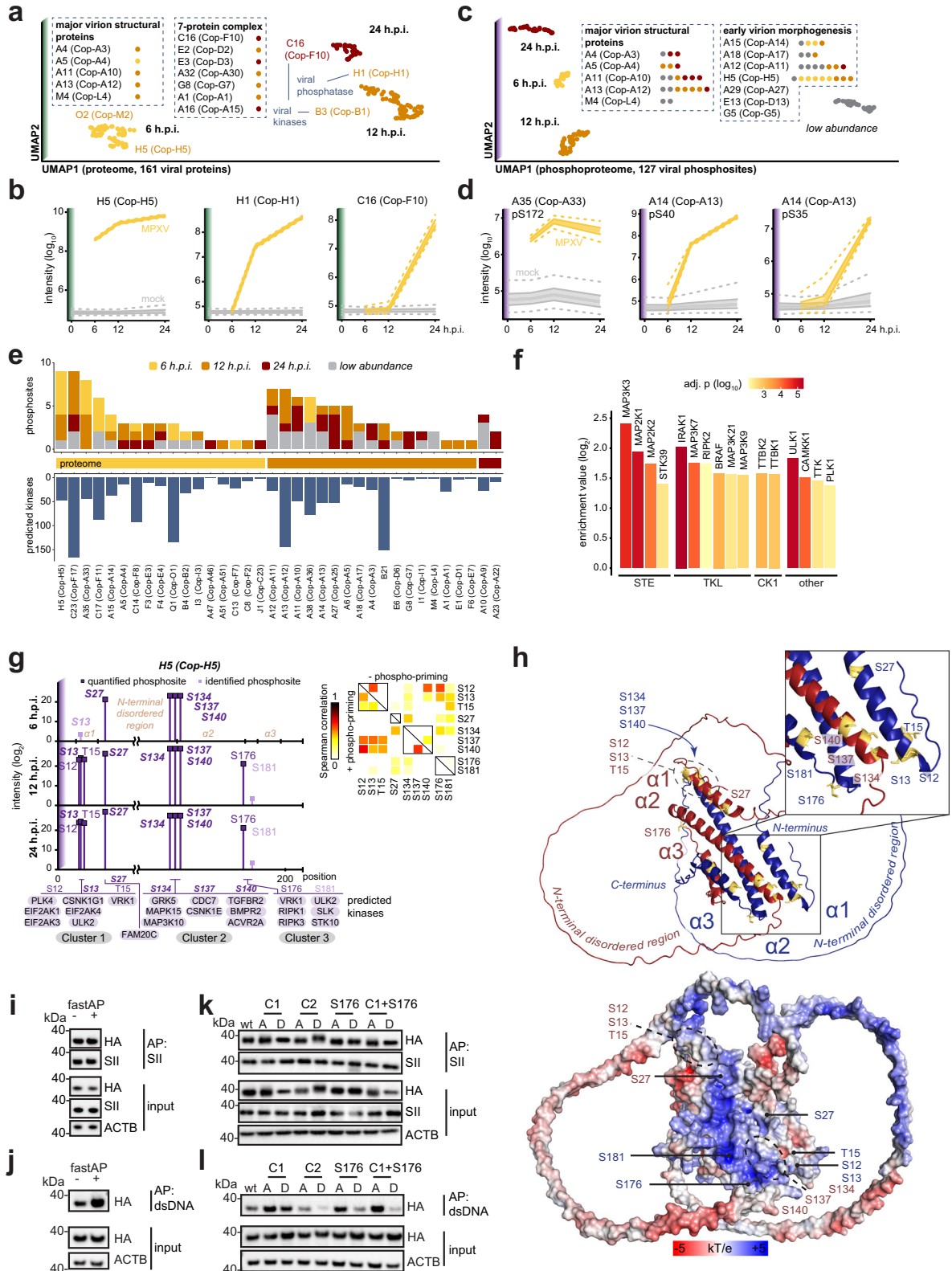

infected cells, H5 is abundantly expressed at 6 hours post-infection (Fig. 2b), with a prominently detectable cluster of phosphosites (S134, S137 and S140, cluster 2) at 6 hours post-infection (Fig. 2g). Interestingly, at 12 hours post-infection, a separate cluster of phosphosites (S12, S13, and T15, cluster 1) begins to appear that further increases in abundance at 24 hours. Similar phosphorylation dynamics could be observed for S176 (Fig. 2g). To gain further insights into the H5

phospho-regulation, we performed multi-chain prediction of the MPXV H5 dimer using AlphaFold[61] and mapped the identified phosphosites on the predicted atomic structure (Fig. 2h, top). The H5 monomers were composed of 3 alpha helices: a shorter N-terminal α1 followed by an expected long N-terminal disordered region connecting it to a longer α2 helix, which is itself connected to C-terminal α3 by a short disordered linker (Fig. 2g, h, top). α1 contains three late

**Fig. 2 | Viral protein dynamics in the multi-omics analysis of MPXV-infected HFFs. a-e** Classification of the viral proteins and phosphosites according to their temporal kinetics. **a** Two-dimensional UMAP of the viral protein abundances in proteomes of HFFs infected with MPXV. **b** Examples of viral proteins. Line: modeled median, shaded region and dotted line: 50% and 95% credible intervals, respectively (*n* = 5 independent experiments). **c** Similar to (**a**), but phosphosite information was used. **d** Similar to (**b**) but phosphosites on viral proteins. **e** Viral phosphosites (top) on individual viral proteins (middle). Bottom: numbers of host kinases, motifs[84] identified at the phosphosites of viral proteins (log2 score > 95% site percentile). **f** Enriched host kinases at the viral phosphosites relative to all detected phosphosites (log2 enrichment score > 0, FDR-adj. p ≤ 0.01 in Fisher's exact tests). X-axis: groups of kinases. **g** Phosphosites detected on the viral protein H5 along the top host kinase motifs. Sites also identified in VACV[27,55] are in bold italics. Right: Spearman rank correlations between host kinases with recognition motifs at individual phosphosites (with or without phospho-priming). **h** Top: In silico predicted structure of the MPXV H5 dimer by AlphaFold[61]. Detected phosphosites are highlighted in gold. Bottom: Electrostatic surface potential analysis of non-phosphorylated MPXV H5 dimers (top). **i** Cell lysates of HEK293T cells transfected with HA- and SII-tagged H5 were left untreated or treated with phosphatase (fastAP) and used for SII affinity purification (AP), followed by western blotting. **j** HA-H5 expressing cell lysates were treated as described in (i) and used for AP using dsDNA as bait. **k, l** The serines (S) or threonins (T) in cluster 1(S12, S13, T15; C1), cluster 2 (S134, S137, S140; C2), cluster 3 (S176), or cluster 1 + S176 were mutated into alanine (A) or aspartic acid (D) in HA-tagged H5, and their binding to SII-H5 **k** or to dsDNA **l** was tested. SII: StrepII tag. *n* = 3 independent experiments for **i-l**. Source data are provided as a Source Data file.

phosphosites (S12, S13 and T15, cluster 1). When phosphorylated, their cumulative negative charge may impair the interaction with the C-terminal α2 from the neighboring subunit that harbors the three early phosphosites (S134, S137, and S140, cluster 2) (Fig. 2h, top). Alongside phosphorylation at S176 and S181 (cluster 3), this may affect self-association, allowing for a potential shift of interaction partners through the liberation of the N-terminal disordered region and, ultimately, a change in activity in the later stages of the virus life cycle. Surface charge prediction models revealed highly positively charged surfaces in the structured part of the H5 dimer (Fig. 2h, bottom), and phosphorylation of residues within or near this area would contribute negative charges, which could impair dsDNA binding. To test whether the phosphorylation of H5 may affect oligomerization or dsDNA binding, we co-expressed HA- and Strep-II (SII)-tagged H5 and left the lysates untreated or treated them with phosphatase (fastAP) to unspecifically remove phosphates. Phosphatase treatment did not affect co-precipitation of HA with SII-tagged H5, indicating that multimerization is not affected by the phosphorylation status of the protein (Fig. 2i). Notably, however, precipitation of HA-H5 with dsDNA was markedly increased in phosphatase-treated lysates (Fig. 2j). To investigate this in more detail, we mutated the identified phosphorylated clusters into phosphoablative alanines (A) or phosphomimetic aspartic acids (D). All mutant proteins co-precipitated similarly with wildtype (wt) H5 (Fig. 2k), confirming that the phosphorylation status of the identified amino acids does not affect the multimerization of the protein (Fig. 2i). In contrast, mutating phosphosites in cluster 1 and S176 to alanines led to better association to dsDNA as compared to wt H5. Moreover, mutation of phosphosites in cluster 2 or concomitant mutation of cluster 1 and S176 into aspartic acids reduced binding to dsDNA (Fig. 2l). Collectively, these data suggest that the phosphorylation status of the here identified residues of H5 affects its interaction with dsDNA and thus the dynamic role of this protein during the viral life cycle. Notably, in VACV H5 displays similar phosphorylation sites on the α2 helix (S127, S130, S133), α1 helix (T11, S13, S27) and in the c-terminal domain of the protein (S183, S191)[27,55] (Supplementary data 4). The complex behavior of viral proteins on the proteomics and phosphoproteomics levels indicated intimate interactions between the virus and the host. To further explore this aspect, we focused on systems analysis of host pathways perturbed by MPXV infection.

## Systems analysis of host response perturbations

To better understand the virus biology and to identify potentially druggable hot spots that the virus relies on, we performed an in-depth systems analysis by projecting the multi-omics data onto the known host biology.

To find pathways that are distinctly or commonly perturbed on multiple signaling levels and potentially play a role in the virus life cycle, we employed an integrative pathway enrichment analysis. Notably, we show that crucial cellular processes are affected at different omics layers with distinct temporal profiles (Fig. 3a, b, Supplementary data 6). For instance, the term peptide chain elongation is enriched among the upregulated transcripts 6 h.p.i., but upregulation of proteins from this process was only detected at 24 h.p.i. (Supplementary data 6). Complex I biogenesis is also enriched among the early upregulated transcripts, but the corresponding protein upregulation was negligible. This was different from previous reports on VACV, where the proteins of OXPHOS were upregulated translationally, but not transcriptionally, during the infection[62]. Moreover, we observed rapid and expected dysregulation of Rho-GTPase signal transduction events in the phosphoproteome analysis starting at 6 hours post-infection, which is gradually followed up by perturbation of the same process on the proteome level[63] (Fig. 3a, b). Furthermore, we observed concerted dysregulation of glycosylation, a process critical for multiple stages of the VACV life cycle[64,65], early upon infection that got more prominent over time on both transcriptional and proteome levels. We additionally identified previously described poxvirus-induced dysregulation of processes on multiple levels, exemplified by downregulation of the collagens[28] and perturbations to the pre-mRNA splicing[50]. In contrast, we also observed significant perturbations in our analysis contained within distinct signaling levels. Such behavior is exemplified by the keratinization on the transcriptome level, inhibition of TSC complex formation by PKB on the proteome level, and the HIPPO pathway on the phosphoproteome level (Fig. 3a, b).

To identify potential host and restriction factors engaged during MPXV infection, we further performed transcription factor enrichment analysis (Fig. 3b, Supp. Fig. 3a, Supplementary data 6). Based on our transcriptomics data, this highlighted differentially active transcription factors associated with the regulation of mRNA abundances. Amongst others, we could identify the expected upregulation of a transcriptional signature regulated by the Early Growth Response 1 (EGR-1)[40]. EGR-1 is a regulator of VACV infection[66] and is particularly interesting as it enables crosstalk to relevant poxvirus responses such as the MEK/ERK pathway[67,68]. Another example specific to the upregulated transcripts is MEF2A, which can be activated by p38[69] and is in line with the detected upregulation of the activating phosphorylation on p38 T180/Y182 (Fig. 1k, l). We also detected an intriguing temporal pattern of the FOSL2 transcript signature. FOSL2 is an AP-1 transcription factor subunit which is involved in several DNA virus life cycles[70–72].

The upstream regulator enrichment analysis of the proteome expression data revealed, among others, the notable involvement of pathways that are regulated by EGF, VEGF, TGF-β, interferon, and integrin signaling in MPXV infection (Fig. 3b, Supp. Fig. 3b, Supplementary data 6). Indeed, poxvirus infections are known to activate EGF-signaling to promote virus spread within the host[73,74]. Interestingly, in our dataset, TGF-β is a key upstream regulator for over 100 proteins significantly dysregulated upon MPXV infection. Several of these proteins have been reported to affect poxvirus growth. Amongst these are RACK1, which is part of the ribosome machinery and phosphorylated by VACV to increase translation[75] (Supp. Fig. 2e), and the VACV restriction factors EPH receptor B2 and DAD1[75,76]. Moreover, we

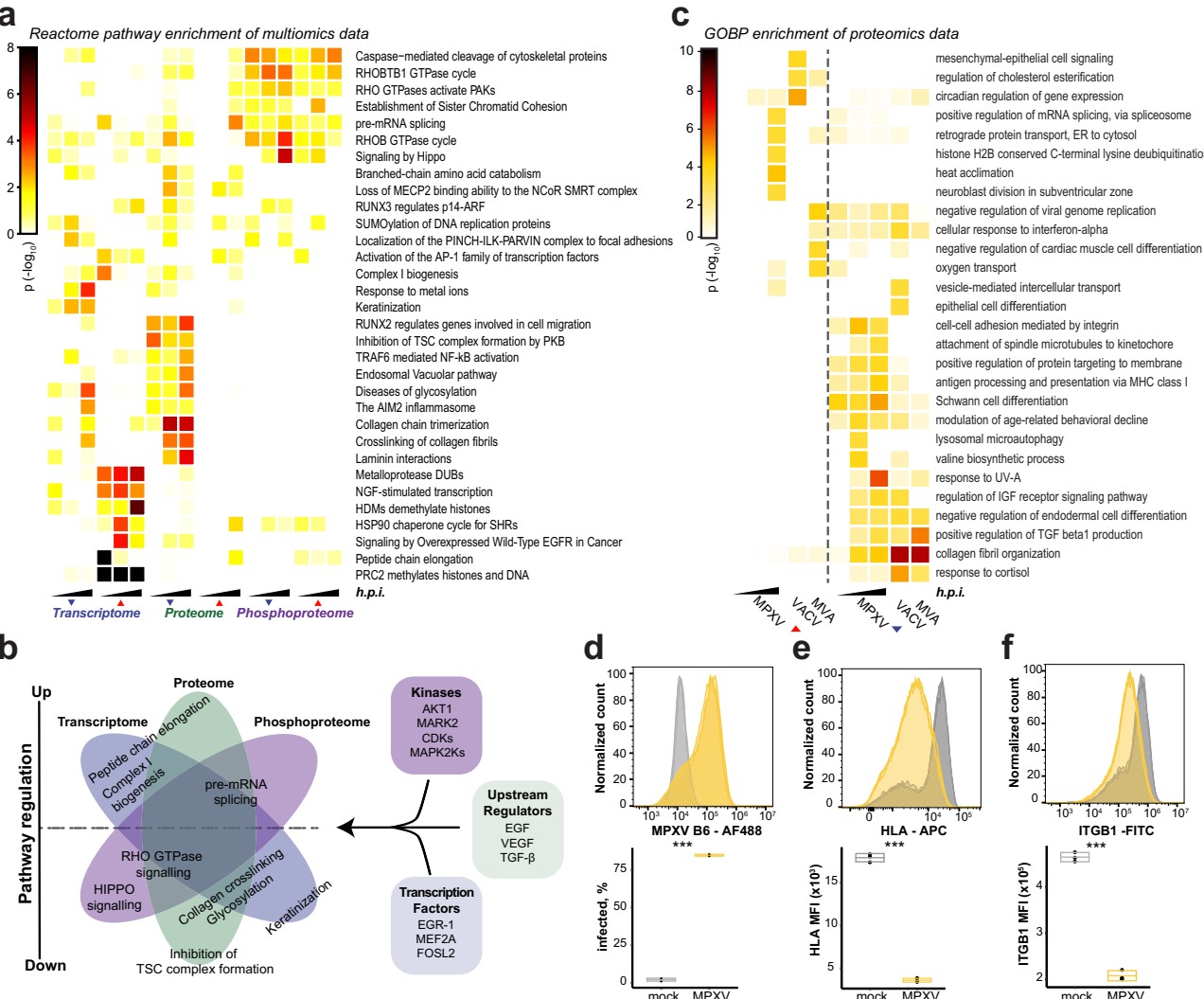

**Fig. 3 | Systems analyzes of the MPXV multi-omics dataset. a** The significant hits from individual layers of the multi-omics dataset were used in an integrative pathway enrichment analysis. Significantly enriched Reactome pathways (Fisher's exact tests, unadjusted $p \leq 0.05$ in at least two conditions) are depicted. **b** Pathways that are differentially activated in MPXV infected cells in different omics layers, as well as a subset of relevant modulators, extracted from Supp. Fig. 3 (a–c). **c** Gene set enrichment analysis of significantly changed proteins in MPXV (this study), VACV[28], and MVA[30]. Graph shows GOBP terms (Fisher's exact tests, unadjusted $p < 0.001$). **d** Intracellular FACS analysis of MPXV B6 protein in HFF cells that were left uninfected or infected with MPXV (MOI: 3) for 24 h. The histograms show fluorescence intensity, and the box plot shows the percentage of infected cells as defined by Supp. Fig. 3e. **e, f** As (d) but cells were analyzed for cell surface abundance of HLA (**e**) and ITGB1 (**f**). The histograms show fluorescence intensity, and the box plots show the median fluorescence intensities of the indicated protein. **d–f** The bounds of box represent lower quartile (Q1) and upper quartile (Q3), and center bar represents median (Q2). ***: $p$-value $\leq 0.001$ ($n = 3$ independent experiments, unpaired two-sided student t-test). Source data are provided as a Source Data file.

detected a subset of heat-shock proteins among the TGF-β regulated proteins, which may be relevant for poxvirus assembly as previously described for HSP90[77,78]. Another potentially relevant contribution of these HSPs could be the synthesis of steroid hormone receptors (SHRs)[79], which were identified across several Reactome pathways (Fig. 3a). Interestingly, it has been shown that VACV suppresses the inflammatory response by hijacking the steroid hormone synthesis[80].

We applied a kinase substrate enrichment analysis based on known kinase-substrate annotations to identify kinases that are active in MPXV-infected cells and that regulate observed changes in the phosphoproteome (Fig. 3b, Supp. Fig. 3c, Supplementary data 6). Here, we could see that phosphorylation patterns on CDKs' substrates were reduced after MPXV infection. This aligns with previous findings in VACV and other poxviruses, which arrest cell cycle progression to favor their own propagation[31,81]. In agreement with the reduced protein abundance of AKT1 after MPXV infection (Fig. 1e), AKT1-regulated sites

were less phosphorylated over the course of the infection, e.g., IRS1 pS629, PDCD4 pS457. In contrast, although no change in protein abundance was observed for MAP2K2/3/4, the host protein phosphorylation pattern indicated their activation in the course of MPXV infection. Interestingly, Cop-O1 of VACV enhances ERK1/2 activation to promote virus replication[82,83] - the sequence conservation with MPXV Q1 (Supplementary data 1) and the detected phosphorylation pattern suggest a conserved activity between these poxviruses. Besides kinases that have been characterized to be relevant for VACV, we also identified increased activity of kinases that have not yet been linked to poxviruses, such as MARK2, a regulator of microtubule dynamics and organization. We further explored the involvement of host kinases using an orthogonal kinase enrichment analysis based on experimentally determined substrate specificities of human S/T kinome[84]. This analysis corroborated the changes in the activity of infection-associated kinases and additionally revealed an orthogonal set of

involved kinases (e.g., ATM/ATR, SRPK, MTOR, DYRKs, etc.) (Supp. Fig. 3d).

Having identified multiple cellular pathways dysregulated in MPXV-infected HFF cells using label-free LC-MS/MS quantification, we compared potential similarities to infections with other poxviruses. Towards this, we considered how protein expression patterns differ between MPXV and VACV[28] and MVA[30] infected fibroblasts, which were analyzed using a tandem mass tag (TMT) based protein quantification by mass spectrometry (Figs. 1d, 3c). We identified similar and unique biological processes that were enriched in these studies (Fig. 3c, Supplementary data 3 and 7). Infection with all viruses resulted in the downregulation of proteins involved in collagen fibril organization and elastic fiber assembly. Similar to MPXV, VACV, and MVA down-regulated proteins involved in interferon or antiviral responses, but MVA additionally upregulated a small subset of ISGs, as expected[44,85,86]. In addition, MPXV and VACV both downregulated HLA molecules[28,87]. Indeed, we further confirmed the downregulation of HLA surface expression in MPXV infected cells by flow cytometry (Fig. 3d, e, Supp. Fig. 3e), which would affect cross talk to NK and T-cells and thus the adaptive immune response. Interestingly, our analysis revealed that MPXV infection disrupted the attachment of spindle microtubules to the kinetochore, a key process during mitosis whose participating proteins were not significantly altered in the VACV and MVA studies (Supplementary data 3 and 7). Poxviruses are known to inhibit the cell cycle at different stages[31,81], and our analysis suggested that MPXV may have additional strategies to interfere with mitosis. Moreover, we observed the downregulation of cell-cell adhesion molecules, including several integrins (ITGA4, ITGA5, ITGB1) and AKT in MPXV infection, which were not significantly regulated in the proteomic studies on VACV and MVA. ITGB1 plays an important role in VACV entry by activating PI3K/Akt pathway[88], and it can form a heterodimer with ITGA4 and ITGA5, respectively. We validated ITGB1 downregulation by flow cytometry and confirmed significant downregulation of this protein in MPXV-infected cells (Fig. 3d, f, Supp. Fig. 3e).

Collectively, our analyzes enabled us to identify regulators of perturbations occurring on transcriptomics, proteomics, and phosphoproteomics levels. Such regulators may directly affect MPXV replication or coordinate the expression of host and restriction factors. Furthermore, the identification of poxvirus-relevant signaling regulators and their downstream targets indicates that our analyzes reveal yet unstudied and potentially druggable host factors.

## Data-driven antiviral drug target prediction

To investigate opportunities for host-directed antiviral drug repurposing, we performed network diffusion-based prediction of potential anti-MPXV drug targets engaged by preclinically and clinically used drugs (Fig. 4a). We first mapped the detected molecular changes from individual omics layers on top of a graph-based representation of host signaling cascades. In particular, we used an expanded multiscale interactome, a graph-based compilation of proteins, GO-terms, diseases, and drugs, connected according to protein-protein interactions, GO-term-, disease- and drug-protein associations[89] (Fig. 4a). This allowed us to associate the infection-elicited molecular fingerprints with potential anti-MPXV drug targets. Random walk with restart was used to propagate the information across the graph-based representation of host signaling, allowing us to statistically estimate the drug-to-virus-infection association (Supplementary data 8, materials and methods). Notably, the drug and drug target predictions from different omics layers were highly orthogonal (Fig. 4b), which reflects the observed patterns in infection-induced pathway perturbations (Fig. 3a) and further underlines the necessity of multi-omics approaches to capture a comprehensive spectrum of drugs and drug targets. Collectively, we identified 1063 drugs and 695 drug targets that were significantly associated with the MPXV fingerprint on the transcriptome, proteome, or phosphoproteome levels (Fig. 4c, d). For

instance, batimastat, the first matrix metalloproteinase inhibitor to reach clinical trials, was significantly associated with infection-elicited molecular cues from proteome profiling (Fig. 4e). Moreover, Torin 2, an experimental drug targeting MTOR, was predicted based on findings from phosphoproteome analysis (Supplementary data 8) and is in line with previous reports on antiviral activity of Torin 2 against VACV[90]. Notably, the pathway enrichment for drug targets reveals a broad diversity of targetable pathways and, again, a high degree of orthogonality between predictions from distinct omics profiles (Fig. 4f, Supplementary data 9). Collectively, this state-of-the-art graph-based prediction approach expanded the findings from biology-oriented integrative analyzes (Fig. 3a, b, Supp. Fig. 3a–d) and provided additional insights on features of the host signaling landscape that can be leveraged for antiviral strategies.

## Drug target assessment and screening

Combining the drug-pathway associations with network diffusion-based drug predictions, we selected 52 drugs (47 small molecules and 5 cytokines) with diverse molecular modes of action, as evaluated by projecting their activities on the mechanism of action data[91], and used them in a proof-of-concept drug target validation screen (Fig. 5a, Supplementary data 10). We performed a live-cell microscopy-based antiviral assay evaluating the ability of selected drugs to attenuate MPXV infection-induced cytopathic effects (CPE). In short, we measured the growth of HFFs in the contexts of treatment, MPXV infection, and the combination thereof, and we employed statistical modeling to determine which drugs protect cells from infection-induced CPE. From the tested drugs, we identified 15 with significant antiviral effects, 6 of which also inhibited cell growth in the uninfected conditions (Fig. 5b, d, Supp. Fig. 4a). Among the non-cytotoxic ones were inhibitors targeting vesicular trafficking (Torin 2, Bafilomycin), receptor tyrosine kinases (RTK) signaling (Regorafenib), pro-inflammatory signaling (ACHP), extracellular matrix regulators (Batimastat), cytoskeleton organization (Fostamatinib), cell proliferation (Binimetinib), DNA damage (VE821), and RNA splicing (SPHINX31). The strongest antiviral activity was observed for Torin 2 (inhibitor of mTOR), ACHP (IKK complex inhibitor), Fostamatinib (Syk inhibitor), Regorafenib (pan-RTK inhibitor), and SPHINX31 (SRPK1 inhibitor). We further tested the ability of the set of drugs to inhibit the growth of a recombinant Vaccinia- GFP reporter virus (Fig. 5c, e, Supp. Fig. 1b). Notably, we observed that all drugs that were antiviral against MPXV were also significantly antiviral against VACV, with the exception of BAPTA AM. Many other drugs that were at the edge of significance for antiviral effect against MPXV were significantly antiviral against VACV-GFP, likely due to an increased dynamic range of this more sensitive assay.

We used orthogonal methods to further corroborate the antiviral effects of the selected subset of compounds against MPXV. Using an RT-qPCR-based assay, we validated that all tested compounds significantly reduced the accumulation of MPXV mRNA, with the strongest dose-dependent effects observed for Torin 2, ACHP, and Fostamatinib (Fig. 5f). Additionally, we quantified the release of infectious viral particles into the supernatants of treated and MPXV-infected HFFs (Fig. 5g). Interestingly, while Tecovirimat completely abrogated production of infectious virus particles, it had only a moderate effect in preventing virus-induced CPE or attenuating viral RNA accumulation upon infection at high MOIs (Fig. 5g). This may be explained by its molecular mode of action, which prevents virion formation and egress[92]. Similarly, the amounts of released virus progeny were also below the detection limit upon treatment of HFFs with Torin 2 and ACHP (Fig. 5g), further demonstrating their potent antiviral efficacy against MPXV (Fig. 5h). Taken together, these findings underline the utility of the gathered and presented orthogonal omics survey of MPXV infection of primary human fibroblasts in elucidating novel virus biology and facilitating drug repurposing.

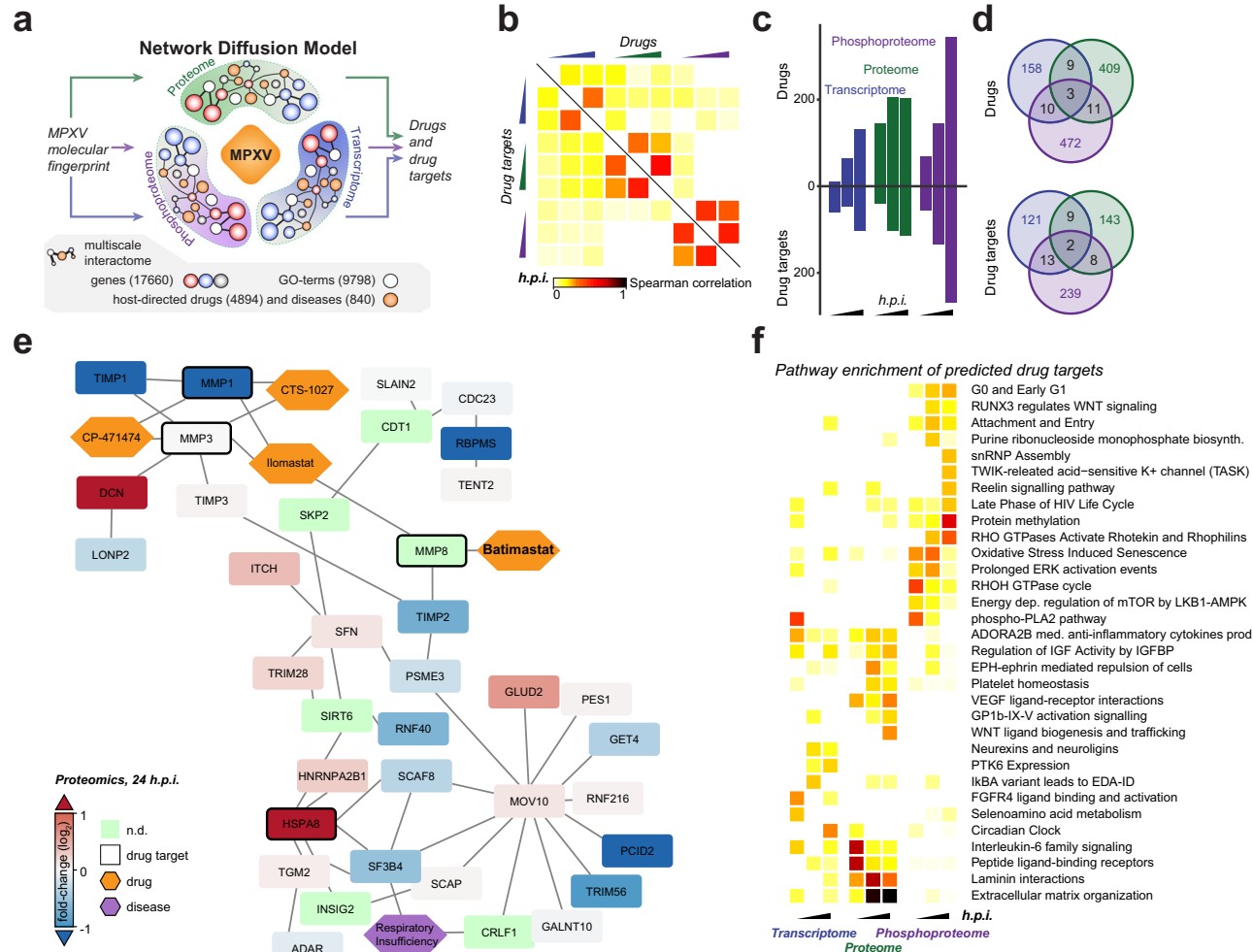

**Fig. 4 | Data-driven prediction of anti-MPXV drugs. a-e** A network diffusion-based host-directed drug repurposing pipeline was used to predict drugs and drug targets associated with the MPXV infection-elicited molecular fingerprints at individual time points of distinct omics layers. **a** Schematic representation of the drug and drug target prediction pipeline based on network diffusion. Multiscale interactome[89] expanded with an additional set of drugs obtained from the drug repurposing hub[108] was used for all graph-based analyzes and predictions (see methods). **b** Spearman rank correlation of drugs and drug targets, ranked according to the network diffusion-based $p$-values from individual time points of the measured omics layers, is depicted as a measure of predictions' orthogonality.

**c, d** Numbers **c** and the overlap **d** of distinct drugs (top) and drug targets (bottom) significantly associated with individual time points of the measured omics layers according to the network diffusion model. **e** The local biological neighborhood of Batimastat and its drug target MMP8, predicted as significantly associated with the MPXV infection-elicited molecular fingerprint on the level of proteome at 24 hours post-infection. **f** The predicted drug targets were used in an integrative pathway enrichment analysis (similar to (Fig. 3a)). Significantly enriched Reactome pathways (Fisher's exact tests, unadjusted $p < 0.01$) are depicted. Source data are provided as a Source Data file.

## Discussion

Herein, we provided a multi-omics study of human primary cells infected with a clinical isolate of the clade II MPXV that caused worldwide outbreaks in 2022 (Fig. 1). This unique dataset allowed us to infer the temporal dynamics of expression and phosphorylation patterns of both host (Figs. 1, 3) and viral proteins (Fig. 2) and shed light on the host-pathogen interactions across multiple regulatory layers, which we validated in our drug screen (Figs. 4, 5).

Notably, despite the extensive literature on the susceptibility of in vivo models to MPXV[93,94], no widely accessible animal model recapitulates well the disease progression observed in patients. Disregarding drug toxicity studies, proof of concept antiviral efficacy studies may be pursued in immunocompromised mouse strains such as Stat1 deficient C57BL/6 mice before proceeding to the young rabbit model[93,94]. Notably, the route of infection and the nature of the MPXV isolate used are expected to have a major impact on any employed model, collectively requiring preclinical efficacy studies to include nonhuman primates such as macaques and marmosets[95].

We observed MPXV infection-induced regulation of numerous critical pathways for various aspects of the virus life cycle and host homeostasis. Poxviruses are particularly notable examples of viruses that extensively reshape both the infected host and its extracellular environment towards a pro-replicative state for the virus[28]. For instance, we showed that MPXV, similar to other poxviruses, heavily impacts cellular homeostasis by hijacking cytoskeleton organization (WASP/PAKs/Rho-GTPases) to facilitate intracellular virion transport[96–101] and EGFR/VEGF signaling to boost cell motility and local spread of infection[74]. Additionally, MPXV perturbs major cell-to-cell signaling hubs such as TGF-β[102] and integrin[88] cascades that collectively utilize central cellular signaling processes (e.g., MAPKs). We further show that MPXV extensively modulates the components of the extracellular matrix, such as collagens and perturbs protease-antiprotease balance through SERPINs, MMPs, and TIMPs (Fig. 1e, h–j, Supp. Fig. 1d–f). Moreover, we show that MPXV impacts the host cell metabolism by dysregulating MTOR and the translation machinery (e.g., RACK1) to divert host resources towards biosynthesis of its own

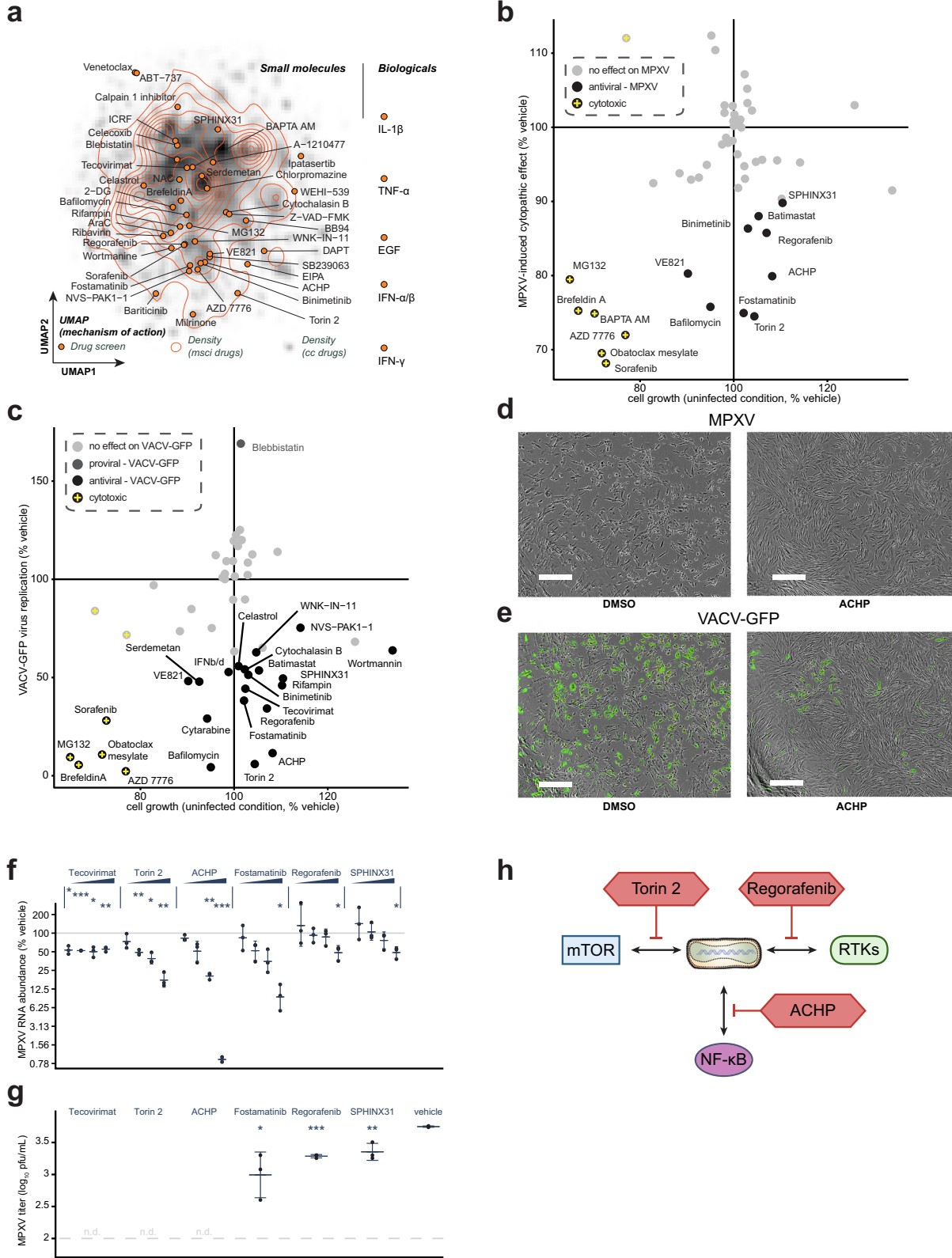

components. At the same time, MPXV promotes host cell survival by dysregulation of the HIPPO/YAP pathway[103] and counter-regulation of cell death induction and related inflammation in response of the host towards infection. We revealed that all of these instances are perturbed by MPXV across one or more signaling modalities (Fig. 3a, b), demonstrating the synergistic utilization of the individual omics profiles in our multi-omics approach.

Our time-resolved multi-omics profiling of MPXV infection revealed interesting temporal regulation of not only the abundances of viral proteins but also their phosphorylation at various sites, which may reflect their involvement at distinct stages of the virus life cycle. Similar to previous reports for VACV[28], many viral antagonists of the innate immune system were detected early upon infection (e.g., A41 (Cop-A41), D11 (Cop-C6), C1 (Cop-K1), O2 (Cop-M2), B9 (Cop-B8)),

**Fig. 5 | Identification and testing of MPXV and VACV inhibitors. a** UMAP of small molecule drugs used in the proof of concept drug screens - the projection was performed using the mechanism of action data obtained from ChemicalsChecker[91]. Densities of drugs in the expanded multiscale interactome (contour lines) and across the ChemicalsChecker database (shades of gray) are depicted for comparison. **b, c** Antiviral assays were performed on hTERT-HFFs, pre-treated with indicated compounds 4 hours before infection with MPXV (MOI 1) **b** or VACV-GFP (MOI 0.1) **c** for 24 hours. Scatterplots depict the drug-dependent reduction in MPXV-infection-induced CPE **b** or VACV-GFP reporter virus growth **c** *versus* the growth rate of cells in uninfected conditions. All values are expressed as percentages relative to vehicle controls. Dark gray/black spots represent significantly effective drugs (linear model-based unadjusted two-sided $p$-value < 0.05; see methods), and the yellow cross indicates the drug is cytotoxic (average relative cell growth <0.75 after drug treatment in uninfected conditions). Multiple concentrations were tested for each drug, and Supplementary data 10 indicates the plotted concentration. **d, e** hTERT-HFF cells treated with DMSO or ACHP (5 μM) for 4 hours and infected

with MPXV (**d**) or VACV-GFP (**e**). The images were obtained 24 hours after the infection. Scale bar = 400 μm. $n$ = 3 independent experiments. **f** hTERT-HFF cells were pre-treated for 4 hours with indicated compounds at increasing concentrations (see methods) and infected with MPXV (MOI 1) for 24 hours. Expression levels of MPXV *G2R* mRNA relative to the housekeeper control *RPLPO* are depicted relative to vehicle (DMSO) controls as measured by RT–qPCR. **g** Infectious viral titers in the supernatants of hTERT-HFF cells from (**f**), which were treated with the indicated compounds at 1 μM concentration, as quantified by plaque assay. **f, g** Data are represented as mean +/- sd from 3 independent biological replicates. Statistical analysis was performed using a two-sided paired student t-test for (**f**), and two-sided unpaired student t-test for (**g**). **h** Antiviral drugs potentially targeting pathways identified in our study. The color indicates the inhibitors used (red) and the putative pathway targeted to restrict MPXV. Pfu: plaque forming unit. *: $p$-value ≤ 0.05; **: $p$-value ≤ 0.01; ***: $p$-value ≤ 0.001. Source data are provided as a Source Data file.

which is in agreement with the observed tight suppression of the host's antiviral defenses. In contrast to this, we identified all members of the seven-protein complex as intermediate/late proteins. This may be related to the interdependence of individual members for stability and the involvement of this complex in immature virion formation at a later stage of the viral life cycle[104,105]. The temporal kinetics of other, less characterized, viral proteins may suggest their potential roles in processes that are active at distinct stages of the viral life cycle (Fig. 2a). Moreover, phosphorylation (Fig. 2c) may alter the function of viral proteins, as suggested by the predicted structural proximity of the H5 (Cop-H5) phosphosites and our experimental data on H5 self-association and H5-dsDNA interaction (Fig. 2g–l). This additional regulation of protein functions could stem from the interplay between the viral and host kinases and phosphatases (Fig. 2e, f). In line with this notion, we observed phosphorylation events on the host and viral proteins previously linked to the activity of viral kinases (Supp. Fig. 2d, e). Moreover, we found enrichment of MAP kinase motifs on identified viral phosphosites (Fig. 2f, Supp. Fig. 2g), which interestingly suggested a direct regulatory role of the MAPK pathways in the virus life cycle. Collectively, our analyzes offer new insights into viral protein regulatory patterns and the cross-talk between viral phosphosites and host kinases.

Our multi-omics analysis of MPXV highlighted many dysregulated pathways as well as potential host factors, some of which are novel while others overlap with what was previously reported for other poxviruses (Fig. 1). Our integrative pathway and regulator analyzes allowed for the identification of diverse host signaling hubs that are potentially important for the virus life cycle (Fig. 3). Notably, the identification and characterization of viral proteins' and phosphosites' dynamics may provide essential information required for the development of novel direct-acting antivirals (Fig. 2). Moreover, we showcased the integration of infection-elicited molecular patterns into our drug repurposing pipeline predicting potential targets of host-directed antivirals (Figs. 3, 4). While further improvements of prediction models that use complex omics datasets are still required to increase their accuracy and facilitate mechanistic interpretation of their results, we collectively demonstrated – on the basis of MPXV infection – that data orthogonality between the omics layers is reflected in orthogonality of drug target predictions originating from them. This underlines the value of multiomics datasets for modeling approaches with broad coverage across the spectrum of host signaling pathways, perturbations of which may best be captured by specific omics layers. This is especially important in the contexts of virus infections, whereby correlations between omics layers, such as between proteomics and transcriptomics, is often profoundly perturbed by viral activities such as host-shutoff. We validated our approaches by demonstrating the successful inhibition of MPXV

replication through targeting the splicing machinery (SPHINX31), mTOR signaling (Torin 2), or the NF-kB response (ACHP) (Fig. 5). Collectively, the herein-presented advancements in our understanding of poxviruses have direct implications for applied antiviral research.

## Methods

### Cell lines and reagents

HEK293T (CRL-11268), Vero E6 (CRL-1586), and BSC40 (CRL-2761) cells were purchased from ATCC. HFF-1 (SCRC-1041) and hTERT-BJ1 cells were a kind gift from Prof. Melanie Brinkmann (HZI, Braunschweig, Germany). Generation of the hTERT-BJ1-H2B-mRFP cell line was done by lentiviral transduction of hTERT-BJ1 with pHIV-H2BmRFP (Addgene #18982) followed by FACS sorting of mRFP-positive cells.

All cells were cultured in DMEM medium (ThermoFisher) supplemented with 10% (vol/vol) FCS (Sigma-Aldrich) and antibiotics (penicillin (100 U/ml) and streptomycin (100 μg/ml)). Total RNA isolation for next-generation sequencing and RT-qPCR was performed according to the manufacturer's protocol (Qiagen RNeasy mini kit, RNase-free DNase set) in 4 replicates, and RT-qPCR was performed as previously described[23]. Primer sequences are provided in Supplementary data 2. For protein abundance detection via western blot, antibodies against MPXV C19 (Cop-F13, a gift from Michael Way, Francis Crick Institute, 1:8000), MMP14 (Abcam, ab51074, 1:2000), PTGS2 (Cell Signaling, 12282, 1:1000), LMNA (Abcam, ab26300, 1:500), THBS1 (Invitrogen, PA5-85678, 1:1000), DAB2 (Cell Signaling, 12906, 1:1000), CTNNB1 (Sigma-Aldrich, C7207, 1:1000), pCTNNB1-S552 (Cell Signaling, 9566, 1:1000), P38 MAPK (Cell Signaling, 8690, 1:1000), pP38 MAPK - T180/Y182 (Cell Signaling, 4511, 1:1000), HA (coupled to horseradish peroxidase (HRP), Sigma-Aldrich, H6533, 1:2500), Strep-Tag™ II (coupled to HRP, Sigma-Aldrich, 71591, 1:4000) and ACTB (β-actin) (coupled to HRP, Santa Cruz, sc-47778, 1:2500) were used. Secondary antibodies against mouse (Cell Signaling, 7076, 1:1000) and rabbit (Dako, P0448, 1:2500) IgG were coupled to HRP. For affinity purification–western blotting applications, Streptactin beads (IBA Lifesciences) were used. The western blot was performed as described before[23]. All compounds used in the virus inhibition assay can be found in Supplementary data 10.

### Virus stock preparation and plaque assay

MPXV was produced in BSC-40 cells and purified from cytoplasmic lysates, as described previously[106]. VACV-V300-GFP was a kind gift from Dr. Joachim Bugert (Bundeswehr Institute of Microbiology, Munich, Germany) and was propagated on Vero E6 cells for three days (MOI 0.001). VACV-V300-GFP infected cells were scraped, sonicated and resuspended in OPTI-MEM before storage at −80 °C. A plaque assay was performed to determine the viral stock titer, as previously done[23]. Briefly, confluent monolayers of Vero E6 cells were infected

with serial fivefold dilutions of virus stock for 1 h at 37 °C before the medium exchange with serum-free MEM (Gibco, Life Technologies) containing 0.5% carboxymethylcellulose (Sigma-Aldrich). Three days post-infection, cells were fixed for 20 min at room temperature with formaldehyde added directly to the medium to a final concentration of 5%. Fixed cells were washed extensively with PBS before staining with water containing 1% crystal violet and 10% ethanol for 20 min. After rinsing with PBS, the number of plaques was counted, and the virus titer was calculated. All work involving MPXV has been conducted according to BSL3 environment safety standards. All virus stocks and cells used for analyzes presented herein were tested to be mycoplasma free.

### HFF proteome and phosphoproteome MS sample preparation

For each of the five replicates, 2 million HFF cells were infected with MPXV (MOI 3). The samples were harvested in SDC buffer (4% SDC, 100 mM Tris-HCl pH 8.5), heat-inactivated (95 °C, 10 min) and sonicated (4 °C, 15 min, 30 s on/30 s off, high settings) at the designated timepoints (0 h, 6 h, 12 h, 24 h). Afterwards, the samples were processed as previously described[107]. In short, protein concentrations were estimated by BCA assay (Pierce) according to the manufacturer's instructions. To reduce and alkylate proteins, samples were incubated for 5 min at 45 °C with TCEP (10 mM) and CAA (40 mM). For each sample, 200 µg protein material was digested overnight at 37 °C using trypsin (1:100 w/w, enzyme/protein, Promega) and LysC (1:100 w/w, enzyme/protein, Wako).

For proteome analysis, 20 µg of peptide material was desalted using SDB-RPS StageTips (Empore). Samples were diluted with 1% trifluoroacetic acid (TFA) in isopropanol to a final volume of 200 µl and loaded onto StageTips, subsequently washed with 1% TFA in isopropanol and 0.2% TFA/ 5% acetonitrile (ACN). Peptides were eluted with 1.25% ammonium hydroxide (NH$_4$OH) in 60% ACN and dried using a SpeedVac centrifuge (Eppendorf, Concentrator Plus). They were resuspended in 0.1 % formic acid (FA) prior to LC-MS/MS analysis. Peptide concentrations were measured optically at 280 nm (Nanodrop 2000, Thermo Scientific) and subsequently equalized using 0.1 % FA.

The rest of the samples were processed according to the published EasyPhos protocol[107]. The samples were diluted 1.3 fold with isopropanol and mixed with 48% TFA in 8 mM Potassium dihydrogen phosphate. Titanium dioxide (TiO2) beads (GL Sciences) were equilibrated in 6% TFA in 80% ACN and 2.5 mg beads were added to each sample. After incubation at 40 °C for 5 min, beads were washed four times with 5% TFA in 60% isopropanol, resuspended in 0.1% TFA in 60% isopropanol and transferred onto C8 StageTips (Empore). Phosphopeptides were eluted from the dry StageTips with 5% NH$_4$OH in 40% ACN. Phosphopeptide sample eluates were diluted with 1% TFA in isopropanol and desalted using SDB-RPS StageTips as described above. Dry peptides were resuspended in 0.1 % FA prior to LC-MS/MS analysis.

### Off-line basic reverse phase fractionation

For offline fractionation 200 µg peptide digest from HFF cells were reconstituted in 0.1 % FA and desalted using Sep-Pak C18 50 mg sorbent cartridges. Peptides were eluted with 0.1% FA in 50% ACN and dried in a SpeedVac. Samples were stored at -80 °C until further use. A Dionex Ultimate 3000 HPLC system (Thermo Fisher Scientific) operating a Waters XBridge BEH C18 3.5 µm 2.1 × 250 mm column was used to fractionate peptides at a flow rate of 200 µl/min with a linear gradient from 0% to 30% buffer B in 45, followed by a linear gradient from 30% B to 89% B (buffer A: 25 mM ammonium bicarbonate pH = 8.0, 5% ACN; buffer B: 25 mM ammonium bicarbonate pH = 8.0, 95% ACN). 200 µl fractions were collected in a 96-well plate and subsequently pooled into 24 fractions. Peptide fractions were frozen at -80 °C and dried in a SpeedVac. Dry peptides were reconstituted in 0.1% FA prior to LC-MS/MS analysis.

### LC-MS/MS data acquisition

HFF proteome and phosphoproteome samples were measured on an Eclipse mass spectrometer (Thermo Fisher Scientific) coupled on-line to a Dionex Ultimate 3000 RSLCnano system (Thermo Fisher Scientific). The liquid chromatography setup consisted of a 75 µm x 2 cm trap column and a 75 µm x 40 cm analytical column, packed in-house with Reprosil Pur ODS-3 3 µm particles (Dr. Maisch GmbH). Peptides were loaded onto the trap column using 0.1% FA in water at a flow rate of 5 µL/min and separated using a 110 min linear gradient from 4% to 32% of solvent B (0.1% (v/v) formic acid, 5% (v/v) DMSO in acetonitrile) (full proteome, HFF fractions) or an 80 min stepped gradient: 4–15% (50 min), 15–27% (30 min) (phosphoproteome) at 300 nL/min flow rate. nanoLC solvent A was 0.1% (v/v) formic acid and 5% (v/v) DMSO in HPLC-grade water. The Eclipse was operated in data-dependent (DDA) and positive ionization mode. Full scan MS1 spectra were recorded in the Orbitrap from 360 to 1300 m/z at 60 k resolution using an automatic gain control (AGC) target value of 100% and a maximum injection time (maxIT) of 50 msec. The cycle time was set to 2 sec. Orbitrap readout MS2 scans were performed using higher energy collision-induced dissociation (HCD) and a normalized collision energy of 30%. For full proteome analysis, the precursor isolation window was set to 1.3 m/z with 15 k MS2 resolution, an automatic gain control target value of 200% and a max IT of 22 msec. For phosphoproteome analysis, the precursor isolation window was set to 1.3 m/z with 30 k MS2 resolution, an automatic gain control target value of 200% and a max IT of 54 msec. Only precursors with charge state 2 to 6 were selected and the dynamic exclusion was set to 45 sec or 35 sec, respectively.

### Processing of raw MS data

All raw MS data files were processed via MaxQuant (version 2.0.3.1) with specific settings stated below. Afterwards, the protein groups were redefined by in-house Julia scripts (protregroup.jl from: https://doi.org/10.5281/zenodo.7757309, dependencies from: https://doi.org/10.5281/zenodo.7752673). In brief, protein groups distinguished by only one specific peptide or had less than 25% different specific peptides were merged to extend the set of peptides used for protein group quantitation and reduce the protein isoform-specific changes.

**HFF proteome MS data**. Raw MS data files of the experiments conducted in DDA mode were processed with MaxQuant (version 2.0.3.1.) using the default settings and label-free quantification (LFQ) (LFQ min ratio count 2, normalization type classic) and intensity Based Absolute Quantification (iBAQ) enabled (protein FDR = 0.01, PSM FDR = 0.01, site FDR = 0.01, max. Missed trypsin site = 2). Spectra were searched against forward and reverse sequences of the reviewed human proteome including isoforms (Uniprot, Taxon ID *9606*) and Monkeypox virus proteins (GenBank: ON563414.3), which shared the highest sequence identity with our clinical isolate, by the built-in Andromeda search engine. The MPXV FASTA was modified to be compatible with Maxquant. The raw data was analyzed using the match between runs option. To increase the sequencing depth an additional dataset of fractionated MPXV-infected cell lysate was used. While this does not affect the analyzed samples, the fractionated MPXV lysates, which were generated using an independent virus isolate, were later on found to be Mycoplasma positive.

**HFF phosphoproteome MS data**. Raw MS data files of phosphoproteome experiments conducted in DDA mode were processed with MaxQuant (version 2.0.3.1.) using the default setting (protein FDR = 0.01, PSM FDR = 0.01, Site FDR = 0.01, max. Missed trypsin site = 2) with the following changes. The parameters "Phospho (STY)" was enabled as a variable modification and "Match between runs" was activated. Spectra were searched against forward and reverse sequences of the reviewed human proteome including isoforms

(Uniprot, Taxon ID *9606*) and Monkeypox virus (GenBank: ON563414.3) proteins by the built-in Andromeda search engine. In total we could identify 14420 phosphopeptides. The PTM was included if it was not marked reverse or contaminant, and detected at least once with localization probability ≥0.75 and posterior error probability (PEP) ≤ $10^{-3}$.

### Bioinformatic analysis
The following bioinformatic analysis was done with R (version 4.1), Julia (version 1.6) and Python (version 3.10) with a set of in-house scripts (https://doi.org/10.5281/zenodo.7757309, https://doi.org/10.5281/zenodo.7752673).

**Datasets.** The following public datasets were used in the study: Gene Ontology and Reactome annotations (http://download.baderlab.org/EM_Genesets/December_01_2021/Human/UniProt/Human_GO_AllPathways_with_GO_iea_December_01_2021_UniProt.gmt); multiscale interactome[89]; drug repurposing hub (https://clue.io/repurposing#download-data)[89,108]; Phosphositeplus (v2022.08)[51,84]; Human (v2021.04) protein sequences from UniProt: https://uniprot.org; MPXV: ON563414.3 protein sequences from GenBank.

**Statistical analysis of MS data.** Due to the scale of this study, which required us to compare perturbations in a time-resolved manner and to deal with sparsity of data as well as statistical noise, we opted to employ an advanced MS-centric data analysis pipeline.

The MaxQuant output files were imported into R with the in-house msimportr R package (https://doi.org/10.5281/zenodo.7746897), which formats the evidence.txt, peptides.txt and proteinGroups.txt tables. The raw and unfiltered data was then analyzed with the msglm package (https://doi.org/10.5281/zenodo.7752068) as described before[23]. The msglm package implements Bayesian linear random effects models and uses the cmdstanr package (version 0.4.0) to infer the protein abundance change between different conditions. Due to the sparsity of the true abundance change caused by experimental conditions, the effects associated with experimental conditions have regularized horseshoe+ priors[109].

The probability of quantifying a peptide increases with its intensity, thus we first evaluated the measurement error of our MS instrument by fitting a heteroscedastic intensity noise model, assuming that quanted intensities follow a mixture of Gaussian and Cauchy distributions. This model was calibrated using technical replicate measurements of the MS instrument. We then used the same data to calibrate a logit-based model of missing MS data, estimating the likelihood of having missing data given the true intensity of an object. Msglm thus handles both quantified and missing values, as both influence the posterior distribution of model parameters, and does not rely on imputation.

The model was fitted with unnormalized MS1 intensities of protein group/PTM-specific peaks (evidence.txt table of MaxQuant output) together with normalization multipliers to better account for the signal-to-noise variation among samples. The normalization multipliers were inferred based on 500 randomly selected MS1 intensities, the assumption being that these selected intensities should be constant between samples. The normalization is hierarchical, shifts were first calculated between the biological replicates within each condition, then between different conditions within the same time point, and finally between different time points. For each individual MS sample, the shifts of each layer are added to obtain a unique normalization multiplier. The sampling for the posterior distribution of model parameters was done via 4000 iterations (2000 warmup + 2000 sampling) of the no-U-turn Markov Chain Monte Carlo in 8 independent chains. The *p*-value was calculated as the probability that two random samples, each from the respective posterior distributions of two different conditions, are different. There was no correction for multiple hypothesis testing because this was resolved via the choice of model priors.

**Statistical analysis of proteomic data of MPXV-infected HFF cells.** The model for HFF infection proteome can be represented by R GLM formula language as

$$log(Intensity(t)) \sim 1 + \sum_{t_i \leq t}(after(t_i) + virus : after(t_i)) + MS1peak, \quad (1)$$

where after($t_i$) effect represents the protein abundance change in mock condition between $t_{i-1}$ and $t_i$ and is included in the model of all time points since $t_i$; virus:after($t_i$) represents the MPXV-specific infection effect between $t_{i-1}$ and $t_i$. MS1peak is the log ratio of an MS1 peak intensity and the total protein abundance as described before[23]. The peak represents a peptide with a specific sequence, PTMs and charge, and the log ratio is assumed to be constant regardless of the experimental conditions[110].

For any comparison between infected and mock samples to be valid, we required the protein group to be quantified in at least 50% of the replicates on either side of the comparison. A significant change at any given time was defined by |median $log_2$ fold-change| ≥ 0.5 and *p*-value ≤ $10^{-2}$.

**Statistical analysis of phosphoproteomic data of MPXV-infected HFF cells.** We applied the same model and definition for valid comparison and significant change as in HFF proteome analysis to the phosphoproteomic data. In addition, we excluded the sites if the protein and phosphosites were both significantly changing in the same direction for the host proteins. For the same phosphosite, different multiplicities were analyzed separately, and the phosphopeptides of the same phosphosite were grouped solely on their different multiplicities without regard to the location of other phosphosites on the peptides. Only the most significant result among all the multiplicities was reported.

**Transcriptomic analysis of MPXV-infected HFF cells.** For the transcriptomic data, Gencode gene annotations v38 and the human reference genome GRCh38 were derived from the Gencode homepage (EMBL-EBI). The MPXV genome was derived from GenBank: ON563414.3. Dropseq tools v1.12 was used for mapping raw sequencing data to the reference genome[111]. Data normalization, differential expression analysis and *p*-value adjustment were performed by the DESeq2 package (version 1.34.0)[112] with the standard setting using the following linear model in R GLM formula language for the four biological replicates:

$$log_2(geneexpression(t)) \sim \sum_{t_i \leq t}(after(t_i) + virus : after(t_i)) \quad (2)$$

with the same effect definitions as in the proteomic analysis. The $log_2$ fold changes were then shrunken via ashr[113]. A valid comparison was defined as when the mean of normalized count on either side of the comparison was at least 40. A gene was considered significantly changing in the valid comparison to mock if adjusted *p*-value ≤ $10^{-5}$ and |shrunken $log_2$ fold-change| ≥ 0.25 (6 h.p.i.), 0.5 (12 h.p.i.) or 1.5 (24 h.p.i.). The different $log_2$ fold-change thresholds were decided based on the difference in the median of absolute fold-change values at the three time points after infection.

**Literature intersection.** For the intersection of full proteome data (VACV[28], MVA[30]), the "sensitive" criteria used by the authors (|fold change | >2) was applied to define significantly changing proteins from the respective studies for the intersection and the following gene set enrichment analysis. The significantly changing proteins from this

study were defined as mentioned in "Statistical analysis of proteomic data of MPXV-infected HFF cells".

For the intersection of phosphosites from VACV[27,55], all the detected phosphosites on viral proteins as listed by the authors in the corresponding supplementary tables were used.

Due to unavailable analysis results in previous transcriptomic studies, we were not able to do a systematic intersection but have listed the relevant studies in Supplementary Table 2 to facilitate potential data mining.

**Gene set enrichment analysis.** The significantly changing genes and proteins defined by the above analysis were used for the integrative pathway enrichment analysis against the Gene Ontology and Reactome databases. The known kinase-substrate annotations were extracted from PhosphoSitePlus[51]. To find the optimal set of annotations that covers the largest amount of significant genes with the least pairwise overlaps, we used the in-house Julia package OptEnrichedSetCover.jl (https://doi.org/10.5281/zenodo.4536596, detailed description of the methods: https://alyst.github.io/OptEnrichedSetCover.jl/stable) as previously described[23]. We define the terms to be significant when the unadjusted Fisher's exact $p$-value ≤ 0.001 for the proteomic literature intersection (Fig. 3c) kinase-substrate analysis of the MPXV phosphoproteome (Supp. Fig. 3c), and $p$-value ≤ 0.05 across at least two conditions for the MPXV multi-omics dataset (Fig. 3a). There was no need for the classical multiple hypothesis testing correction due to the high selectivity of the algorithm.

**Transcription Factor Enrichment Analysis.** For transcription factor enrichment analysis (Supp. Fig. 3a) the significantly regulated transcripts were submitted to ChEA3 web-based application and ENCODE data on transcription factor–target gene associations were used[114,115]. Transcription factors were considered as significant if they exceeded a FDR-adjusted $p$-value threshold of 0.001 according to Fisher's exact tests.

**Upstream regulator analysis.** For the identification of global upstream regulators, significant hits from the full proteome analysis were processed in the ingenuity pathway analysis software (version 84978992, Qiagen). The core analysis was performed using the default settings including the Ingenuity Knowledge Base as the reference set for $p$-value calculations as well as "direct and indirect relationships" for upstream regulator analysis. After analysis, only upstream regulators with an unadjusted $p$-value < 0.05 were considered significant. For visualization (Supp. Fig. 3b), upstream regulators belonging to molecule types *growth factor*, *transmembrane receptor* or *group* were shown.

**Viral protein and phosphosite kinetics.** For the analysis of temporal abundance patterns of viral proteins (Fig. 2a) and phosphosites (Fig. 2c), the model estimated median changes between infected and the time point matched mock conditions for either viral proteins or phosphosites were used. We performed uniform manifold approximation and projection (UMAP) dimensionality reduction in R (4.0.2) using R package UMAP[116] (0.2.6.0) and manually annotated thus obtained clusters.

**Prediction of kinases for detected phosphomotifs.** The prediction of host kinases that phosphorylates detected phosphosites on the host (Supp. Fig. 3d) and viral proteins (Fig. 2e–g, Supp. Fig. 2f–h) was performed by using the Kinase Library toolbox[84] (kinase-library.phosphosite.org).

The enrichment analysis for viral phosphosites (Fig. 2f) was performed using all detected viral phosphosites as foreground. For host phosphosites (Supp. Fig. 3d), the significantly changing sites at individual time points, as defined according to the previous section, were used as a foreground. All analyzes used all detected phosphosites as background. The host kinases with a positive log2 enrichment score and FDR-adjusted $p$-value ≤ 0.01 according to one-sided Fisher's exact tests in at least one condition were displayed in the respective figures.

Site-wise prediction for phosphosites of viral proteins was performed using a single site scoring algorithm without (all viral phosphosites, Fig. 2e, Supp. Fig. 2f, g) or with optional inclusion (H5 and A14, Fig. 2g and Supp. Fig. 2h) of phospho-priming as indicated. In the latter case, the phosphopeptide sequence was modified so that any detected phosphosites in the 5 amino acid vicinity of the analyzed phosphosite were considered phosphorylated. For Fig. 2e and Supp. Fig. 2f–g, kinases with a positive log2 score above 95% site percentile were counted. Top predictions are shown for Fig. 2g and Supp. Fig. 2H. The host kinases were filtered (percentile > 95% in any of the shown predictions) prior to the calculation of the Spearman rank correlations (Fig. 2g, Supp. Fig. 2H).

**H5 structure modeling with AlphaFold and electrostatic surface potential analysis.** In silico prediction of the structure of MPXV H5 dimer was performed using the colab version of AlphaFold[61] 2.3.1 in the multichain mode using default parameters. Electrostatic surface potential of the modeled structure of MPXV H5 dimer was calculated by using the PyMOL plugin APBS electrostatics. Molecular graphics depictions were produced with the PyMOL software.

**Network diffusion analysis.** All subsequent analyzes were based on the graph-based representation built from the multiscale interactome[89] (disregarding GO-term hierarchy) with the addition of drug-to-drug target relationships obtained from the drug repurposing hub[108]. In total, the resulting graph contained the following heterogeneous nodes: drugs (4894), diseases (840), genes (17660) and GO-terms (9798). The graph was constructed in an undirected manner with the exception of drug-to-drug target (genes) edges and disease-to-disease-related gene edges, which were made unidirectional and directed towards the drugs/diseases. Hyperparameters, i.e. edge weights and restart probability, were taken from the multiscale interactome[89]. Random walk with restart kernel (R) was computed according to the following equation: R = alpha * (I − (1−alpha)*W)$^{-1}$, where I is the identity matrix, and W is the weight matrix computed as W = D$^{-1}$ * A, where D is degree diagonal matrix, and A is adjacency matrix for the above-constructed graph. The diagonal values of the R matrix, representing restart and feedback flows, were excluded from subsequent analysis and set to 0.

The significant hits from individual time-points of omics analyzes were mapped to genes by matching gene names or, if that failed, their synonyms (from the biomaRt_hsapiens gene ensembl dataset) to the gene names in the multiscale interactome. In the transcriptomics dataset, histones were excluded from the analysis. Nodes with significant sum of inbound network diffusion flows originating from nodes representing hits in individual analyzes were estimated using a randomization-based approach. All hits and non-hits of the analysis were attributed equal weight (1 and 0, respectively). Flows to all nodes in the network were computed by multiplying the R matrix with the vector of hit weights as described above. Furthermore, all nodes in the network were assigned to 8 bins of approximately equal size according to the node degree. The same procedure of calculating inbound flows to all network nodes was repeated for 10,000 iterations, each time using the same number of randomly selected decoy hits according to degree binning. The $P$-values describing the significance of functional connectivity to input hits were computed for each node according to the following formula: $P$ = N(iterations with equal or higher inbound flux as real hits)/N(iterations). The following cutoffs were used: p(gene nodes) <0.01 (0.001 and 0.005 for phosphoproteome at 24 and 12 h.p.i., respectively), p(other) <0.01. For a drug to be considered significant, at least one of its drug targets also had to reach significance

levels as described above. Analogous additional criterion was also applied to diseases and GO terms. Drug target pathway enrichment analysis (Fig. 4f) was performed as described above (section Gene set enrichment analysis), with pathways with a $p$-value < 0.01 considered significant.

For visualization purposes (Fig. 4e), the graph nodes were filtered for significance as described above and pairwise semantic similarity between genes was calculated using GOSemSim 2.14.2[117] (based on GO biological processes and Resnik similarity measurement[118]). We clustered graph nodes (genes only, other terms were re-mapped onto thus obtained clusters) based on gene semantic similarities using the method apcluster in the R package apcluster 1.4.8[119] with damping parameter *lam* set to 0.915. Specifically, we gradually increased the input preference parameter ($p$) in steps of 5 from -500 to -20 until the input was split into 2 or more clusters. The clusters were further separated into weakly connected components, trimmed, and the process repeated until reaching sizes below 75 (transcriptomics 6 h.p.i, proteomics 12 and 24 h.p.i., and phosphoproteomics 24 h.p.i) or 50 genes. We performed trimming by removing nodes with a single connection that were not quantified in the dataset used for network diffusion unless they were connected to a significant drug or disease node. For visualization purposes, drugs and diseases with a single connecting edge were also trimmed from the displayed network. With this, we separated the graph-based output of the network diffusion prediction pipeline into local biological neighborhoods approximating pathways to facilitate result visualization and interpretation.

**Drugs' mechanism of action visualization.** For the visualization of drugs used in the drug screen according to their mechanism of action (Fig. 5a), data from ca. 800.000 bioactive small molecules was acquired from ChemicalChecker[91] (chemicalchecker.org) as vector signatures. Uniform manifold approximation and projection (UMAP)[116] was used for dimensionality reduction to 2 displayed dimensions using UMAP's python API v0.5.3. The mapping of drugs from the extended multiscale interactome and drug screen to this dataset was performed by matching of drug names and manual curation, respectively.

**Co-immunoprecipitation and western blot analysis**

For the H5 homodimer co-immunoprecipitation experiment, HEK293T cells were transfected with pCAGGS plasmid encoding single HA-tagged wildtype H5 or its phosphoablative (S/T to A) or phosphomimetic (S/T to D) mutants, together with pCAGGS plasmid encoding single StrepII-tagged wildtype H5. The following phosphosites in each of the three clusters were mutated simultaneously (cluster 1: S12, S13, T15; cluster 2: S134, S137, S140, cluster 3: S176). The sequence of H5 and its mutants were codon-optimized, and potential splice sites were removed (GeneArt, ThermoFisher). 24 h after transfection, cells were washed in PBS, lysed with lysis buffer (50 mM Tris-HCl pH 7.5, 100 mM NaCl, 1.5 mM MgCl2, 0.2% (v/v) NP-40, 5% (v/v) glycerol, cOmplete protease inhibitor cocktail (Roche), 0.5% (v/v) 750 U µl − 1 Sm DNase) and sonicated (5 min, 4 °C, 30 s on, 30 s off, low settings; Bioruptor, Diagenode SA). The lysates were cleared by centrifugation. For FastAP (ThermoFisher) treatment, the same lysate containing wildtype H5 protein was split into two equal portions of around 300 µg protein each, 30 U FastAP was added to one portion, and both portions were incubated at 37 °C for 1 h. Streptactin beads were added to cleared lysates, and samples were incubated for 2 h at 4 °C under constant rotation. Beads were washed twice in the lysis buffer and twice more in the lysis buffer without NP-40, and resuspended in 1× SDS sample buffer (62.5 mM Tris-HCl pH 6.8, 2% SDS, 10% glycerol, 50 mM DTT, 0.01% bromophenol blue). After boiling for 5 min at 95 °C, a fraction of the input lysate and elution were subjected to western blot analysis.

For the dsDNA-H5 interaction experiment, HEK293T cells were transfected with pCAGGS plasmid encoding single HA-tagged wildtype H5 or its phosphoablative (S/T to A) or phosphomimetic (S/T to D)

mutants alone and after 24 h, lysed in the lysis buffer as mentioned, but without Sm DNase (Benzonase). Double-stranded interferon stimulatory DNA[120] was used as the dsDNA bait and prepared as described before[121]. The dsDNA-bound beads were added to cleared lysate, and samples were incubated overnight at 4 °C under constant rotation. The washes and subsequent western blot analysis were performed as mentioned above.

**Flow cytometry**

For each of the three replicates, 1 million HFF cells were infected with MPXV (MOI 3) or left uninfected for 24 hours. The cells were detached by 5 mM EDTA for subsequent staining. Cell surface staining was performed using anti-pan-HLA-APC (clone W6/32, Biolegend, 1:500) or anti-ITGB1-FITC (clone Ha2/5, BD Pharmingen, 1:500) antibodies. Dead cells were excluded from analysis by Zombie NIR Fixable Viability Dye (Biolegend, 1:1000) staining. Cells with surface staining were fixed in 4% formaldehyde and permeabilized with 0.1% Triton (if used for intracellular staining). Intracellular staining was performed using anti-MPXV B6 (originally anti-VACV B5, a gift from Michael Way, Francis Crick Institute, 1:1000) and donkey-anti-rat-AF488 (ThermoFisher, A21208, 1:1000). Samples were measured on a CytoFlexS flow cytometer (Beckmann Coulter, USA) and analyzed using FlowJo software (v10.9.0, Tree Star, USA).

**Antiviral assays based on live-cell fluorescent imaging**

The antiviral assays were performed in a batch-wise fashion, with each batch including vehicle controls, and were processed and analyzed accordingly. hTERT-BJ1-H2B-mRFP cells were seeded in 96-well plates one day before infection. Four hours before infection, cells were treated with the indicated compounds and concentrations or vehicle controls (DMSO and PBS). Infection was performed by the addition of viral stock, either MPXV (MOI 1) or VACV-V300-GFP (MOI 0.1). 96-well plates were placed in the IncuCyte S3 Live-Cell Analysis System (Essen Bioscience), where images of mock (phase channel) and infected (GFP and phase channel) cells were captured at 0 h and 24 hours post-infection. The ratio between cell confluence at 24 hours *versus* 0 hours was calculated as a measure of cell growth and MPXV- or drug-induced cytotoxicity. For VACV-V300-GFP infected samples, the GFP area normalized to cell confluence (GFP area/phase area) was used as a measure of virus growth. Described data handling was performed using IncuCyte S3 Software (Essen Bioscience; version 2020 C rev1) and exported for further analysis as described below. The whole assay was repeated 3 times.

**Analysis of the drug cytotoxicity.** For each drug treatment in uninfected conditions, we compared the above-described cell growth measure to the respective vehicle. Drugs with a relative cell growth <0.75 were determined to be toxic and marked as such. The highest non-toxic concentration for each drug was included in the below-described modeling approach (Fig. 5b,c) and thus included in the figures. When all tested concentrations were cytotoxic, the lowest concentration was included in the modeling and figures.

**Analysis of the MPXV antiviral drug assay.** We fitted cell growth measurements from mock- and MPXV-infected drug-treated cells using the following linear model:

$$log_2(cell\ growth) \sim 1 + biological\ replicate\ number + infection + drug \\ + infection \bullet drug$$

(3)

In this modeling approach, the *biological replicate number* effect quantifies the variation between biological replicates. The *infection* effect quantifies the MPXV infection-induced CPE on the cells in vehicle-treated conditions. The *drug* effect quantifies the drug's cytotoxicity in the mock-infected conditions. Finally, the *infection · drug*

effect quantifies the drug-dependent reduction or increase in MPXV infection-induced CPE, thereby serving as a measure of drugs' antiviral effects. In Fig. 5b, the *infection · drug* effect is plotted on the y-axis, while the *drug* effect is plotted on the x-axis (also plotted on the x-axis of Fig. 5c). A drug is considered effective when *p*-value < 0.05 for the *infection.drug* effect.

**Analysis of the VACV-GFP antiviral drug assay.** As for the VACV-GFP drug screen, we fitted the virus growth measurements according to the following linear model:

$$log_2(virus\ growth) \sim 1 + biological\ replicate\ number + drug \quad (4)$$

Similar to the MPXV modeling approach, the *biological replicate number* effect quantifies the variation between biological replicates and the *drug* effect quantifies drug-dependent reduction or increase in virus growth. The *drug* effect is plotted in Fig. 5c (y-axis) as a measure of the antiviral activity of the drugs. A drug is considered effective when *p*-value < 0.05 for the *drug* effect.

### Antiviral assays based on RT-qPCR

The validation of the antiviral activity of the selected screened compounds by RT-qPCR was performed in the following manner. hTERT-BJ1-H2B-mRFP cells were pre-treated with ranging concentrations of Tecovirimat, ACHP, Fostamatinib, Regorafenib (0.1, 0.25, 1 or 5 μM) or Torin 2 (0.025, 0.1, 0.25 or 1 μM) for 4 hours prior to infection with MPXV (MOI 1). After 24 hours, supernatants were collected for plaque assay analysis and cells were lysed for RNA extraction (Macherey-Nagel NucleoSpin RNA plus) according to the manufacturer's protocol. cDNA synthesis (PrimeScript RT with a gDNA eraser, Takara) and qPCR analysis (PowerUp SYBR Green, Applied Biosystems) was performed as previously described[23]. RT-qPCR was performed using primers targeting *G2* of MPXV (fw: 5'-GGAAAATGTAAAGACAACGAATACAG-3'; rev: 5'-GCTATCACATAATCTGGAAGCGTA-3') and the human housekeeper *RPLPO* (fw: 5'-GGATCTGCTGCATCTGCTTG-3'; rev: 5'-GCGACCTG-GAAGTCCAACTA-3'). ΔΔCt values were calculated as [Ct(*RPLPO, drug*) - Ct(*G2R, drug*)] - [Ct(*RPLPO, vehicle*) - Ct(*G2R, vehicle*)] (Fig. 5f).

### Reporting summary
Further information on research design is available in the Nature Portfolio Reporting Summary linked to this article.

## Data availability
The raw sequencing data for this study have been deposited with the ENA at EMBL-EBI under accession number PRJEB60728. The files of the proteomic datasets and Maxquant output have been deposited to the ProteomeXchange Consortium (http://proteomecentral.proteomexchange.org) via the PRIDE partner repository. This includes the following datasets: Full Proteome HFF/MPXV (PXD040811), Phosphoproteome HFF/MPXV (PXD040889). Source data are provided with this paper.

## Code availability
In-house R and Julia packages and scripts used for the bioinformatics analysis of the data have been deposited to public GitHub repositories:msglm: https://doi.org/10.5281/zenodo.7752068 msimportr: https://doi.org/10.5281/zenodo.7746897. OptEnrichSetCover: https://doi.org/10.5281/zenodo.4536596. analysis_utils_jl (package dependencies for the julia packages and scripts used in this manuscript): https://doi.org/10.5281/zenodo.7752673 General scripts: https://doi.org/10.5281/zenodo.10685047

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

## Acknowledgements

The work in the author's laboratory was funded by the ERC (ERC-CoG ProDAP, 817798 to APic; VIROFIGHT consortium, 848223 to UP; ENDFLU 874650 to GS and EFRE-REACT to UP and GE), the Federal Ministry of Education and Research (BMBF) (FKZ 01KX2026 and FKZ 01KI20702 to GS and FKZ16LW0286K to GE), SFBs of the German research foundation (TRR179-272983813 to APic and UP, TRR237-369799452 to APic and FIS, TRR353-471011418 to APic and GE), German research foundation (INST 95/1650-1) to APic, Germany's Excellence Strategy (EXC2151-390873048) to FIS, the Center for Immunology of Viral infections (CiViA) funded by the Danish National Research Foundation (DNRF 164) to APic, the Helmholtz Association's Initiative and Networking Fund (KA1-Co-02 "COVIPA" to APic and UP; KA1-Co-06 "CORAERO" to GE), the Volkswagen Foundation (Az.9B825) to GE, Science Foundation Ireland (16/IA/4376 and 22/FFP-A/10459) to AGB. MW is supported by the Francis Crick Institute, which receives its core funding from Cancer Research UK (CC2096), the UK Medical Research Council (CC2096), and the Wellcome Trust (CC2096). We would like to thank Thomas Enghleiter and Niklas de Andrade Kraetzig for their help with the analysis and management of the transcriptomic dataset, Alexey Stukalov and Melissa Verin for their support on data analysis, Stefan Lichtenthaler for offering essential reagents and Paula Deuter for her technical support.

## Author contributions

Y.H., V.B., V.G., Q.E., P.H., J.M., A.Pir., K.K., A.H., R.Ö., and T.B. conducted experiments. Y.H., V.B., V.G., Q.E., M.S.H., and R.Ö. analyzed data. Y.M.T. and I.D.R. contributed critical reagents. Y.H., V.B., V.G., Q.E., M.W., and A.Pic. designed the experiments and wrote the paper. G.S., G.E., F.I.S., M.W., R.R., A.G.B., U.P., and A.Pic acquired funding.

## Funding

## Competing interests

V.G., and A. Pic. are co-inventors on a patent application related to the inhibition of intracellular pathogen uptake by ACHP. The rest of the authors have no competing interests.
