## [Peer Review File · Nature Communications]

Multi-omics characterization of the monkeypox virus infectionREVIEWER COMMENTS

Reviewer #1 (Remarks to the Author):

This is a comprehensive analysis of the transcriptome, proteome and phosphoproteome of MPV infected cells. I am not aware of a similar large scale analysis being published elsewhere and this work is timely and of wide use in the field.

Ultimately the authors show that analysis of the dataset outcomes leads to the implication that there are druggable targets that could affect the outcome of viral infection.

They screen a number of such drugs and a small proportion appear to have promise. I do not have many corrections or observations except:

1. It would be useful to provide a wider context to MPXV infections in the introduction - e.g. what treatment options are available.
2. The potent outcomes of the work are the ability of the drugs to inhibit virus replication so I think supplementary figure 5 should be made a main paper figure.
3. I think Supplementary figure 5 could then be images of one or two other drugs and their effect in cell culture in restricting virus replication (e.g. Torin2 and Fostamantinib)
4. I think the authors need to explore in their discussion how these studies might now be expanded by giving a short overview of the animal models available (including shortcomings of MPXV animal models).
5. Do they have commentary on why some drugs did not affect virus replication?

Reviewer #1 (Remarks on code availability):

I'm not qualified to review the code, but I did look at the README and it appears to be a generic readme file that you are supposed to edit and fill in for your software. They do not appear to have edited the README properly so it is specific to their software.

Reviewer #2 (Remarks to the Author):

In this report, Huang et al. use an array of 'omics and analytical techniques to extensively characterize MPXV infection in human fibroblasts. The study provides impressively detailed insights into the dynamics of infection on multiple levels. A slight negative is that it doesn't uncover very much that isn't already known, but it provides a valuable resource for the MPXV and broader poxvirus community. It also demonstrates how decades of knowledge can be gained from the combined application of 'omics approaches, and how this could be used to accelerate drug discovery in the face of future pandemics caused by less well understood viruses. However, there are some significant concerns around some of the proteomics and comparisons, how the broader 'omics analyses truly integrate and what they truly tell us. For these reasons, I am supportive of this study, but a number of issues would be important to address.

Main Concerns:

While there are broad transcriptomic and proteomic changes that seem similar from a big picture view in figures 1b and 1d, further compared in generalized GO terms in figure 3a, there is a notable lack of strong overlap in many cases. Looking at the supplemental datasets in more detail, there doesn't seem to be a strong overlap between transcript and protein changes on the individual gene level, and a concern is that GO term similarities may be arising from different genes that fall within similar GO term classifications. Some of the proteomic data was hard to examine as it was dominated by viral proteins. It would be helpful to present a separate dataset of just the host proteome and a direct comparison of individual genes where the transcript and corresponding protein change in the same direction, and the relative magnitude of changes. It would also be helpful to present GO terms for these specific gene sets that correspond at both levels. If I am not simply mistaken in my understanding of the data, I think this is actually an important point within the overall dataset as it may point to a high level of post-transcriptional control that has also been suggested for VACV from proteomic studies cited.

It would also be important, and relatively simple, to validate some of the changes by western blotting. This is important because the study does not use quantitative TMT-MS but instead compensates computationally on the back end, which is not as accurate. In addition, some of the specific proteins referred to in the text don't seem to be decreased or increased particularly robustly in the excel datasets. Others are discussed in general terms that are vague or lacking in

specifics. For example, in the paragraph beginning on line 229 the authors point to prior profiling studies that suggested translational upregulation of ribosomal and OXPHOS proteins, but this study only validated two OXPHOS proteins which were barely affected on the protein level and don't appear as hits in MS screens by others, cited here, creating a lot of uncertainty over how well the RNA landscape influences the proteome in infected cells. In the current study, groups of proteins such as peptide elongation factors are said to be increased with a lag that follows earlier increases in transcript levels, but I could not determine which specific factors they were or how much they changed. This relates to the point above, where it would be helpful to show the RNA and Protein correlations for specific genes that are being discussed, and to validate a reasonable number of these decreases and increases by western blotting. Currently, only one protein decrease is confirmed in figure 1i, along with two changes in phosphorylation in figure 1k, which have been shown previously in VACV systems.

Given concerns that transcript data may not be a reliable predictor of changes in protein levels, it seems very possible that its inclusion in the network diffusion model in figure 4 could reduce the accuracy of the predictions. What does the network prediction look like, and compare with the drug predictions in figure 5, if only proteomic and phosphoproteomic data are used, as these measure the direct targets of drugs? It may be more accurate.

The authors point out differences in MPXV 'omics versus reports for VACV strains, WR and MVA. It would be important for the authors to clearly point out that the proteomic approaches used in these studies were very different, particularly because they used more quantitative TMT rather than computational compensation used here. As such, the differences shown in this manuscript are likely to be as much down to technical differences than actual differences between the viruses. I do not think this a major issue as the authors do not make excessive claims about these differences, and concerns that TMT is more accurate would be alleviated if validation is done as suggested above, but I think these major technical differences should be clearly pointed out so that readers do not overinterpret these comparisons themselves.

Minor

There is no supplemental figure 4?

Line 640: What is 2 Mio of cells? Million?

Reviewer #2 (Remarks on code availability):

I am not good enough at coding to review this

Reviewer #3 (Remarks to the Author):

In their manuscript, Huang et al. performed multi-omics analysis of the transcriptome, proteome and phosphoproteome of monkeypox-infected human fibroblasts for three different time points to reveal the virus-host interplay that occurs during infection. They use their data to highlight virus-induced perturbations of host cell processes, reveal temporal regulations of viral protein expression and phosphorylation, and to predict potential drug targets and drugs via integrated pathway analysis.

Strengths: Given the recent outbreaks of monkeypox infection in humans the study is certainly timely and addresses the need to better understand this pathogen in order to develop strategies for combating the infection. The authors collected a very large amount of multi-omics data that they provide in an accessible manner as a resource for the field. Importantly, the described observations do not rely on a single dataset but integrate results from different omics approaches with temporal resolution. The presentation of the data is clear and concise and there is a strong emphasis on linking the data acquired with the published literature on poxvirus biology. In its current form the manuscript would therefore be a highly valuable resource.

Weaknesses: Even though the results that the authors reveal from their data rely on integrated data analysis the conclusions can still be considered predictions, in particular regarding the relevance of the observed changes to pathways. More experimental validation with assays other than the multi-omics approaches for some of the conclusions would strengthen the manuscript. For example, it is quite clear from the drug predictions that even the authors' integrated analysis does not result in high confirmation rates. Only a small fraction of the drugs tested possess antiviral

activity.

Points to address:

- The introduction section could provide more background. What are the differences in disease between VACV and MPXV? Did the authors have a hypothesis which differences in virus-host cell interactions would contribute to disease outcome? Is there evidence that multi-omics approaches have helped develop antiviral drug strategies?
- Fig. 2: The authors' observation of differential phosphorylation of the viral protein H5 is intriguing but the study would benefit from some functional validation. Can the authors determine the interactome of H5 at different times post infection? Or could certain functions of H5 be tested in the presence of specific kinase inhibitors?
- L.318: The authors state that their analysis has revealed restriction factors of MPXV. I could not see candidate restriction factors. Could the authors clarify here which data they refer to?
- Fig. 5: A large set of drugs does not have antiviral activity and the ones that do seem to be stimulating cell growth (5b lower right quadrant). How specific is the antiviral function here in the CPE-based assay? Can the authors show that other cytolitic viruses do not get inhibited?

REVIEWER COMMENTS

Reviewer #1 (Remarks to the Author):

This is a comprehensive analysis of the transcriptome, proteome and phosphoproteome of MPV infected cells. I am not aware of a similar large scale analysis being published elsewhere and this work is timely and of wide use in the field.

Ultimately the authors show that analysis of the dataset outcomes leads to the implication that there are druggable targets that could affect the outcome of viral infection.

They screen a number of such drugs and a small proportion appear to have promise. I do not have many corrections or observations except:

1. It would be useful to provide a wider context to MPXV infections in the introduction - e.g. what treatment options are available.

We thank Reviewer #1 for raising this point. The revised version of the manuscript has an expanded introduction on the phenotype of MPXV vs vaccinia virus, current treatment options and how multi-omics data can benefit the identification of bioactive molecules. We feel that this addition improves the manuscript and thank Reviewer #1 for this suggestion.

2. The potent outcomes of the work are the ability of the drugs to inhibit virus replication so I think supplementary figure 5 should be made a main paper figure.

We thank Reviewer #1 for this comment and moved the Supp. Fig. 5 to main Figure 5.

3. I think Supplementary figure 5 could then be images of one or two other drugs and their effect in cell culture in restricting virus replication (e.g. Torin2 and Fostamantinib)

We added pictures of the infection with or without Torin 2 and Fostamatinib treatment as additional examples to Supplementary Figure 4 (formerly Supplementary Figure 5).

4. I think the authors need to explore in their discussion how these studies might now be expanded by giving a short overview of the animal models available (including shortcomings of MPXV animal models).

We agree with Reviewer #1 and have added an overview of animal models in the discussion (starting from line 437).

5. Do they have commentary on why some drugs did not affect virus replication?

Reviewer #1 is correct in that not all drugs that were predicted using the omics data affected MPXV under the experimental conditions used. However, this may be expected due to the complexity of the primary data used for drug prioritization. The identification of drugs is based on the combined activity of perturbation patterns directly imposed by MPXV and the organism's response to the virus infection. While both the direct virus insult and the cellular

response would factor into the prioritization of drugs, we expected that these drugs would be of functionally distinct activities.

State-of-the-art drug search algorithms, such as the one employed in our manuscript, at this stage do not allow us to more accurately pinpoint drug functionality, and therefore, functional experiments are necessary to define the antiviral activity of the identified drugs. We are currently working towards more fine-grained analysis pipelines that aim at a better functional characterization of drug activity.

Reviewer #1 (Remarks on code availability):

I'm not qualified to review the code, but I did look at the README and it appears to be a generic readme file that you are supposed to edit and fill in for your software. They do not appear to have edited the README properly so it is specific to their software.

We thank Reviewer #1 for pointing it out and have updated the README file with the specific information about the code.

Reviewer #2 (Remarks to the Author):

In this report, Huang et al. use an array of 'omics and analytical techniques to extensively characterize MPXV infection in human fibroblasts. The study provides impressively detailed insights into the dynamics of infection on multiple levels. A slight negative is that it doesn't uncover very much that isn't already known, but it provides a valuable resource for the MPXV and broader poxvirus community. It also demonstrates how decades of knowledge can be gained from the combined application of 'omics approaches, and how this could be used to accelerate drug discovery in the face of future pandemics caused by less well understood viruses. However, there are some significant concerns around some of the proteomics and comparisons, how the broader 'omics analyses truly integrate and what they truly tell us. For these reasons, I am supportive of this study, but a number of issues would be important to address.

Main Concerns:

While there are broad transcriptomic and proteomic changes that seem similar from a big picture view in figures 1b and 1d, further compared in generalized GO terms in figure 3a, there is a notable lack of strong overlap in many cases. Looking at the supplemental datasets in more detail, there doesn't seem to be a strong overlap between transcript and protein changes on the individual gene level, and a concern is that GO term similarities may be arising from different genes that fall within similar GO term classifications.

We thank Reviewer #2 for the detailed analysis of our datasets and the comments raised. Indeed, Reviewer #2 is correct that the correlation between the transcriptomic and proteomic changes in MPXV-infected cells is less than what may be expected in other biological contexts. This may partially be due to post-transcriptional, translational, and post-translational processes, which are known to affect protein expression in infected cells. A classic example is the activation of the NF κ B pathway, where the I κ B α is phosphorylated, ubiquitinated, and degraded, so the changes are detected mainly on the proteomic and PTM levels. The activated NF- κ B will induce the transcription of many inflammatory cytokines, whose changes will be observed in transcriptomic but less prominently in proteomic data due to their release into the extracellular space. Therefore, we expect that changes in a pathway or biological function may manifest as individual gene/protein changes depending on the observed omic layer. However, since pathways combine proteins in functionally linked entities, the analysis of the data on pathway level is one way to identify similarities and discrepancies between different omics datasets.

In the GO enrichment analysis presented in our manuscript, we used a customized algorithm (OptEnrichSetCover; code available and described in GitHub; see code availability) that unbiasedly identifies the most specific GO terms that cover the most detected changes. Moreover, having multiple time points from multiple omics layers improves the accuracy of term selection and offers a comprehensive overview of the altered biological processes during MPXV infection.

Some of the proteomic data was hard to examine as it was dominated by viral proteins. It would be helpful to present a separate dataset of just the host proteome and a direct comparison of individual genes where the transcript and corresponding protein change in the same direction, and the relative magnitude of changes.

We agree with Reviewer #2 that many proteomic and phosphoproteomic changes come from viral proteins and have now included a column “is_viral” in Supplementary Tables 3 and 4 so that the readers can conveniently filter out viral proteins while browsing the data. We have further re-designed Supplementary Table 3 to allow a side-by-side comparison between the transcriptomic and proteomic data, as suggested by Reviewer #2.

It would also be helpful to present GO terms for these specific gene sets that correspond at both levels. If I am not simply mistaken in my understanding of the data, I think this is actually an important point within the overall dataset as it may point to a high level of post-transcriptional control that has also been suggested for VACV from proteomic studies cited.

The transcriptomic and proteomic data are poorly correlated to each other, and we could only identify a limited set of corresponding changes on both the mRNA and protein levels (3, 44, and 77 genes at 6, 12, and 24 h.p.i., respectively). These gene sets were only enriched for “extracellular matrix organization” using the Reactome database at 24 h.p.i. This is expected in virus-infected cells as viruses commonly interfere with the host gene expression, also post-transcriptionally, and different pathways may be mediated at different stages of gene expression. This result further supports our choice of enrichment analysis strategy that considered all changes in transcriptomic, proteomic, and phosphoproteomic data.

It would also be important, and relatively simple, to validate some of the changes by western blotting. This is important because the study does not use quantitative TMT-MS but instead compensates computationally on the back end, which is not as accurate. In addition, some of the specific proteins referred to in the text don't seem to be decreased or increased particularly robustly in the excel datasets. Others are discussed in general terms that are vague or lacking in specifics. For example, in the paragraph beginning on line 229 the authors point to prior profiling studies that suggested translational upregulation of ribosomal and OXPHOS proteins, but this study only validated two OXPHOS proteins which were barely affected on the protein level and don't appear as hits in MS screens by others, cited here, creating a lot of uncertainty over how well the RNA landscape influences the proteome in infected cells. In the current study, groups of proteins such as peptide elongation factors are said to be increased with a lag that follows earlier increases in transcript levels, but I could not determine which specific factors they were or how much they changed. This relates to the point above, where it would be helpful to show the RNA and Protein correlations for specific genes that are being discussed, and to validate a reasonable number of these decreases and increases by western blotting. Currently, only one protein decrease is confirmed in figure 1i, along with two changes in phosphorylation in figure 1k, which have been shown previously in VACV systems.

We agree with Reviewer #2 and now provide additional RT-qPCR and Western blot analyses for dynamically changed genes and proteins to further strengthen the omics datasets (Fig. 1h, j). We confirmed that THBS1 and DAB2 (reduced in both mRNA and protein abundance) and PTGS2 (increased in both mRNA and protein abundance) were altered by MPXV infection underlying transcriptional control of these genes/proteins. Conversely, in MPXV-infected cells, the mRNA abundance of *CTNNB1* and *MMP14* does not correlate with protein abundance. In addition, the protein abundance of LMNA/C remained relatively stable despite the decrease in their mRNA. We have included the relevant figures in Figure 1 and amended the text accordingly in the revised version of the manuscript.

Reviewer #2's comment suggested a lack of clarity when we explained the effects of MPXV on OXPHOS and peptide chain elongation. We now individually report the effects of MPXV infection on peptide chain elongation and OXPHOS, which we feel makes this paragraph easier to understand. Regarding differential regulation of OXPHOS proteins, we compared our results to reported data in Ref 48, which emphasized the translational upregulation of OXPHOS genes during VACV infection. The authors further showed in Fig. 4b that all the OXPHOS genes were downregulated at mRNA level 4-8 h.p.i., and validated that three proteins (MT-CO1, MT-CO2, SDHB) increased their abundance during VACV infection. However, in our study, the majority of OXPHOS gene mRNAs increase in MPXV-infected samples at 6 h.p.i., and the same applies to the ribosomal proteins. This discrepancy between published data and our results may be due to the employed experimental system or may be virus-specific.

In relation to the peptide elongation factors, we, in the revised version of the manuscript, now refer to Supplementary Table 6, which lists the affected proteins in the column "genes_overlap". This will better guide the readers to the data that is relevant to this paragraph.

Additionally, we now provide a side-by-side comparison of the transcriptomic and proteomic data in Supp. Table 3, as mentioned above.

Given concerns that transcript data may not be a reliable predictor of changes in protein levels, it seems very possible that its inclusion in the network diffusion model in figure 4 could reduce the accuracy of the predictions. What does the network prediction look like, and compare with the drug predictions in figure 5, if only proteomic and phosphoproteomic data are used, as these measure the direct targets of drugs? It may be more accurate.

We thank Reviewer #2 for raising this concern. Regarding the predictive power of transcriptomics data in the particular case of MPXV infection, we agree that it is lower than other omics layers despite the fact that it is the most data-rich (12970 quantified genes). This is evident by the low number of significant predictions originating from the transcriptomics data (Fig. 4c and d), especially from the 6h.p.i. time point (Fig. 4b and c). While it is able to predict drug targeting potential for several pathways, which are partially orthogonal to other omics layers, the predictions are less correlated between time points and ultimately less robust than predictions obtained by either proteomics or phosphoproteomics (Fig. 4f).

Nevertheless, as our prediction pipeline (Fig. 4a) considers individual omics layers separately, the sub-par performance of transcriptomics does not impact predictions originating from the other omics layers.

We want to emphasize that the information content of various omics layers depends on the specific biological system tested. The observations made herein for MPXV infection should not be generalized to other contexts. Cellular signaling occurs on numerous molecular levels, and in various contexts such as virus infections, disruption of correlation likely occurs between other layers as well, such as post-translational modifications and proteomics/transcriptomics. This adds a degree of complexity to either manual or algorithm-based interpretation of omics data and is one of the central reasons for the necessity of multi-omics studies.

The authors point out differences in MPXV 'omics versus reports for VACV strains, WR and MVA. It would be important for the authors to clearly point out that the proteomic approaches used in these studies were very different, particularly because they used more quantitative TMT rather than computational compensation used here. As such, the differences shown in this manuscript are likely to be as much down to technical differences than actual differences between the viruses. I do not think this a major issue as the authors do not make excessive claims about these differences, and concerns that TMT is more accurate would be alleviated if validation is done as suggested above, but I think these major technical differences should be clearly pointed out so that readers do not overinterpret these comparisons themselves.

We thank Reviewer #2 for this comment - we added information on the technical differences between the VACV and MVA studies in lines 340ff.

Minor

There is no supplemental figure 4?

We apologize for this, we now show supplemental figures in consecutive order.

Line 640: What is 2 Mio of cells? Million?

We have corrected it in the method, it should be “million” indeed.

Reviewer #2 (Remarks on code availability):

I am not good enough at coding to review this

Reviewer #3 (Remarks to the Author):

In their manuscript, Huang et al. performed multi-omics analysis of the transcriptome, proteome and phosphoproteome of monkeypox-infected human fibroblasts for three different time points to reveal the virus-host interplay that occurs during infection. They use their data to highlight virus-induced perturbations of host cell processes, reveal temporal regulations of viral protein expression and phosphorylation, and to predict potential drug targets and drugs via integrated pathway analysis.

Strengths: Given the recent outbreaks of monkeypox infection in humans the study is certainly timely and addresses the need to better understand this pathogen in order to develop strategies for combating the infection. The authors collected a very large amount of multi-omics data that they provide in an accessible manner as a resource for the field. Importantly, the described observations do not rely on a single dataset but integrate results from different omics approaches with temporal resolution. The presentation of the data is clear and concise and there is a strong emphasis on linking the data acquired with the published literature on poxvirus biology. In its current form the manuscript would therefore be a highly valuable resource.

Weaknesses: Even though the results that the authors reveal from their data rely on integrated data analysis the conclusions can still be considered predictions, in particular regarding the relevance of the observed changes to pathways. More experimental validation with assays other than the multi-omics approaches for some of the conclusions would strengthen the manuscript. For example, it is quite clear from the drug predictions that even the authors' integrated analysis does not result in high confirmation rates. Only a small fraction of the drugs tested possess antiviral activity.

Points to address:

- The introduction section could provide more background. What are the differences in disease between VACV and MPXV? Did the authors have a hypothesis which differences in virus-host cell interactions would contribute to disease outcome? Is there evidence that multi-omics approaches have helped develop antiviral drug strategies?

We thank Reviewer #3 for the positive evaluation and interest in the study. The revised version of the manuscript contains a more expanded introduction, which was indeed very brief in the previously submitted manuscript. We now explicitly mention that the vaccinia virus is originally a vaccine strain with limited pathology as compared to MPXV, which recently caused outbreaks in the human population. We also added additional information on differences in clinical manifestation of both MPXV and vaccinia virus infection. Combined efforts of infection biologists, clinicians, and computational scientists showed the high value of multi-omics data to identify drugs that are active against SARS-CoV-2 (Gordon et al., Comparative host-coronavirus protein interaction networks reveal pan-viral disease mechanisms. *Science*, 2020). Moreover, we and others have extensively employed proteomics analysis to describe pathways that are activated or perturbed by viruses and that

suggest mechanisms contributing to viral pathogenicity. Since MPXV was an enigmatic pathogen, it was not clear at the start of this project which cellular pathways are affected by MPXV infection. This prompted us to generate the dataset presented here, which describes the MPXV infection process on multiple omics layers and will, therefore, be highly useful to fellow scientists. The revised version of the introduction now better explains this rationale and highlights the values of such datasets for individual disciplines.

- Fig. 2: The authors' observation of differential phosphorylation of the viral protein H5 is intriguing but the study would benefit from some functional validation. Can the authors determine the interactome of H5 at different times post infection? Or could certain functions of H5 be tested in the presence of specific kinase inhibitors?

We thank Reviewer #3 for this very interesting suggestion, which prompted us to study the influence of H5 phosphorylation on the activity of this protein in detail. Notably, we found an exciting change in H5's function related to the differential phosphorylation status.

As suggested by Reviewer #3, we tested the homo-dimerization of H5 and the binding of H5 to double-stranded DNA in the presence or absence of phosphatase treatment (new Fig. 2i, j). While the dimerization of H5 was not affected by treatment with phosphatases, association of H5 to ds DNA was clearly improved after phosphatase treatment. This further prompted us to generate a series of phospho-ablated and phospho-mimetic H5 mutants and to test their homo-dimerization and ds DNA binding capability. As for phosphatase treatment, changing the identified phosphorylated residues did not affect the dimerization of H5 (new Fig. 2k). Notably, however, mutating S12/S13/T15 or S176 into alanine markedly improved the association of H5 to ds DNA (new Fig. 2l). In contrast, mutating the same residues into phosphomimetic aspartic acid did not result in strong association.

These data show that efficient dsDNA binding by H5 is facilitated by dephosphorylated residues and that phosphorylation of the newly identified residues leads to a functional change in the activity of this protein. Moreover, the wt H5 binds comparable to dsDNA as the phosphomimetic mutants, while phosphoablative mutants bind stronger. This indicates that the protein is naturally phosphorylated when expressed in trans. We now describe the new results in Figure 2i-l. We feel that the analysis of phosphorylation mutants provides evidence that the identified phosphorylated residues affect the functionality of H5. We thank Reviewer #3 for the suggestion to evaluate the H5 phosphorylation status in detail since this clearly improved the manuscript.

- L.318: The authors state that their analysis has revealed restriction factors of MPXV. I could not see candidate restriction factors. Could the authors clarify here which data they refer to?

We apologize for the misunderstanding. Our systems analysis identified a number of upstream regulators that are likely responsible for the changes we observed in different omics data. Many of these regulators changed abundance or phosphorylation status during MPXV infection, further underlining the involvement of these factors in MPXV-host interactions.

The virus may interact with these regulators in other ways that are not covered by our study, e.g., protein-protein interaction or other PTMs. We ask for Reviewer #3's understanding that studying these interactions would exceed the scope of this manuscript. However, the regulators identified by our analysis may be potential host or restriction factors, which can be tested for their influence on MPXV infection in the future. In the revised version of the manuscript, we have adjusted the sentence to clarify this point.

- Fig. 5: A large set of drugs does not have antiviral activity and the ones that do seem to be stimulating cell growth (5b lower right quadrant). How specific is the antiviral function here in the CPE-based assay? Can the authors show that other cytolytic viruses do not get inhibited?

The identification of potential antiviral drugs based on omics data is still a heavily researched area of research and recent reports, particularly fuelled by the SARS-CoV-2 pandemic, indicate the value of -omics data to prioritize drugs. However, multi-omics data contains multiple facets of virus-host interactions, which include the engagement of virus proteins and nucleic acids with host factors, virus-specific perturbation of the immune system, cellular adaptation to compensate for virus-induced perturbations antiviral countermeasures, and cellular innate immune responses. While both the direct virus insult and the cellular response would contribute to the prioritization of drugs, the functional consequence of the identified small molecules would be expected to have different outcomes.

CPE-based antiviral efficacy assays suffer from false positive discoveries when facing, in our experience, rare, pro-proliferative drugs and false negatives when facing more common cytotoxic or cytostatic agents. For this reason, we opted for validation of CPE-based antiviral assays utilizing RT-qPCR-based viral mRNA quantification as well as traditionally used plaque assays. Both methods, to a large extent, validated the findings of our screen. This is in line with our observations that the scattering of cell-growth stimulating capabilities of drugs that we considered "antiviral" (without negative effects on cell viability) lies within an expected range that would be expected in such experiments. Some drugs that did not affect MPXV-induced cell death potentially stimulated cell growth, further illustrating that there is no correlation between cell growth induction and antiviral activities.

We tested for the influence of the drugs used on cell density, and for a subset of drugs, we also tested whether the antiviral effect was more general or not. In fact, we found that ACHP (an IKK inhibitor) is also antiviral against SARS-CoV-2 and partially antiviral against Herpes Simplex virus 1. However, it is not antiviral against viruses such as Measles and VSV, which also cause cytopathic effects. The specificity of ACHP is a phenomenon that we are currently pursuing in a follow-up project. However, this example illustrates that the identified drugs are partially specific, but the general antiviral activity of the identified drugs cannot be deduced.

REVIEWERS' COMMENTS

Reviewer #2 (Remarks to the Author):

The authors have addressed my concerns although I agree with the limitations pointed out by reviewer 3. Optional, but it may be worth noting the low success rate in predicting drug candidates from such a large scale characterization of infection. This is perhaps unsurprising as the inherent complexity will make predictions challenging, so I support the publication as much for its importance as a resource as its ability to predict drug targets with a high degree of accuracy. But since this is a central claim, it may be worth commenting on this.

Reviewer #2 (Remarks on code availability):

I am not suitable experienced to review the code

Reviewer #3 (Remarks to the Author):

The authors have addressed some of my previous points. I appreciate the revised introduction section and the efforts made to provide functional data on the role of H5 phosphorylation.

However, two important points remain for me:

- Fig. 2i-l: These assays to elucidate the roles of H5 phosphorylation strengthen the manuscript but in their current form they lack a specificity control: There is no sample included that shows that the pull-down is specific and dependent on the used antibody. I am guessing that the authors have done these controls before running the assay but in my opinion, the specificity control should be included in the figure or as a supp. figure.

- Fig. 5: On the one hand the authors "sell" their manuscript as an important step to identify host-directed drugs but when addressing my point about the specificity and the success rate for identifying drugs they argue that it is all very difficult. Given that they make such a strong point about the usefulness of omics datasets for finding drugs they need to address this point better. Testing the hit compounds with activity against MPXV also against VSV is a simple assay that would help the reader assess the specificity of the drugs and thus better judge the outcome of the drug identification.

Reviewer #3 (Remarks on code availability):

I am not qualified to assess the code.

Reviewer #2 (Remarks to the Author):

The authors have addressed my concerns although I agree with the limitations pointed out by reviewer 3. Optional, but it may be worth noting the low success rate in predicting drug candidates from such a large-scale characterization of infection. This is perhaps unsurprising as the inherent complexity will make predictions challenging, so I support the publication as much for its importance as a resource as its ability to predict drug targets with a high degree of accuracy. But since this is a central claim, it may be worth commenting on this.

We thank the reviewer for raising this point. We expanded the discussion on this topic in the last paragraph of the discussion section.

Reviewer #2 (Remarks on code availability):

I am not suitable experienced to review the code

Reviewer #3 (Remarks to the Author):

The authors have addressed some of my previous points. I appreciate the revised introduction section and the efforts made to provide functional data on the role of H5 phosphorylation. However, two important points remain for me:

- Fig. 2i-i: These assays to elucidate the roles of H5 phosphorylation strengthen the manuscript but in their current form they lack a specificity control: There is no sample included that shows that the pull-down is specific and dependent on the used antibody. I am guessing that the authors have done these controls before running the assay but in my opinion, the specificity control should be included in the figure or as a supp. figure.

We now provide a new Supplementary Figure 2, which shows the requested control. We show that HA-H5 does only bind to SII-beads when the beads are coupled with SII-H5 (leading to H5 dimerisation) or with dsDNA (H5 binding to DNA). These results address the points raised by Referee#3 and are indeed a nice control to present in the manuscript.

- Fig. 5: On the one hand the authors “sell” their manuscript as an important step to identify host-directed drugs but when addressing my point about the specificity and the success rate for identifying drugs they argue that it is all very difficult. Given that they make such a strong point about the usefulness of omics datasets for finding drugs they need to address this point better. Testing the hit compounds with activity against MPXV also against VSV is a simple assay that would help the reader assess the specificity of the drugs and thus better judge the outcome of the drug identification.

We thank the reviewer for this comment. We are very excited by the development of tools to identify drugs based on omics data analysis, which was not possible until recently. In this manuscript we further improved this method to implement knowledge on functionalized protein interaction networks. Also considering the comment from Reviewer #2, we now expanded the discussion on this topic in the last paragraph of the discussion section. We agree that the use of predictive models that utilize (multi-)omics datasets towards drug target

assessment is a developing methodology, and current implementations need to be improved both to increase accuracy as well as interpretability of their results. Unfortunately, little can be said about the specificity of this approach when considering a single virus at the time, such as in case of this manuscript. For this reason, addressing specificity by testing activity of the compounds against evolutionary divergent viruses is certainly conceptually interesting, but since we here do not compare omics-based analysis of Mpox with VSV, direct comparison between these two viruses would not address whether the predictions are valuable or not. To address this, one would have to perform comparative omics analysis of both pathogens and evaluate active and inactive drugs. However, although interesting this is beyond the scope of current study.

Reviewer #3 (Remarks on code availability):

I am not qualified to assess the code.